# ACCELERATING ERROR CORRECTION CODE TRANSFORMERS

## ABSTRACT

Error correction codes (ECC) are crucial for ensuring reliable information transmission in communication systems. Choukroun & Wolf (2022b) recently introduced the Error Correction Code Transformer (ECCT), which has demonstrated promising performance across various transmission channels and families of codes. However, its high computational and memory demands limit its practical applications compared to traditional decoding algorithms. Achieving effective quantization of the ECCT presents significant challenges due to its inherently small architecture, since existing, very low-precision quantization techniques often lead to performance degradation in compact neural networks. In this paper, we introduce a novel acceleration method for transformer-based decoders. We first propose a ternary weight quantization method specifically designed for the ECCT, inducing a decoder with multiplication-free linear layers. We present an optimized self-attention mechanism to reduce computational complexity via code-aware multi-heads processing. Finally, we provide positional encoding via the Tanner graph eigendecomposition, enabling a richer representation of the graph connectivity. The approach not only matches or surpasses ECCT's performance but also significantly reduces energy consumption, memory footprint, and computational complexity. Our method brings transformer-based error correction closer to practical implementation in resource-constrained environments, achieving a 90% compression ratio and reducing arithmetic operation energy consumption by at least 224 times on modern hardware.

## 1 INTRODUCTION

Reliable digital communication systems rely heavily on ECC to ensure accurate decoding in the presence of noise. Developing efficient decoding techniques for these codes remains a complex challenge in communications research. In recent years, the application of machine learning to communications has driven the development of advanced decoding methods, leveraging deep learning architectures (Nachmani et al., 2016; 2017; Gruber et al., 2017; Kim et al., 2018; Nachmani & Wolf, 2019; Buchberger et al., 2020; Choukroun & Wolf, 2024a;c). Notably, the work of Choukroun & Wolf (2022b) introduced a Transformer-based decoder (Vaswani et al., 2017) adapted to the ECC setting, demonstrating significant improvements over traditional methods across multiple code families.

Despite these advancements, the ECCT and similar neural decoders face significant challenges due to their high memory requirements, energy consumption, and computational complexity. These resource-intensive solutions pose substantial barriers to deployment in many physical communication systems, where efficiency and practicality are paramount, thus constraining the broader adoption and further refinement of these advanced decoding techniques.

Neural network (NN) quantization offers a promising approach to addressing these challenges. Recent research has shown that constraining NN weights to 1-bit and ternary representations can be effective (Ma et al., 2024; Wang et al., 2023), particularly when combined with 8-bit activations. This approach replaces multiplication operations with integer addition, significantly reducing energy consumption and memory footprint. However, applying extreme quantization techniques to smaller models presents considerable challenges. Wang et al. (2023) demonstrated that while the performance gap between BitNet and FP16 Transformers narrows as model size increases, this gap

is particularly pronounced in smaller models. For instance, a BitNet model with 100M parameters (considered small) showed a 10% higher loss than its full-precision counterpart. This disparity would be even more severe for ECCT, whose largest version contains only 2 million parameters. While Ma et al. (2024) improved upon this method, significant performance gaps remain in smaller models, partly due to the use of absolute mean quantization, which lacks flexibility in dynamically adjusting weight sparsity during training.

Recent research on self-attention mechanisms has focused on reducing complexity and memory usage, particularly in large language models. Two main approaches have emerged: sparse attention methods (Beltagy et al., 2020; Zaheer et al., 2020; Child et al., 2019) and attention approximations (Choromanski et al., 2020). However, these techniques were not designed to optimize smaller models, such as ECCT, which are also more sensitive to the information loss that occurs when applying sparse attention or self-attention approximations than larger ones, due to the limited number of layers. In addition to the capacity-related challenges, ECCT's unique architecture poses additional ones. ECCT's inherently sparse code-aware mask is incompatible with sparse attention methods, since it cannot be reduced further without modifying the information brought by the code. Similarly, attention approximation methods are incompatible because they bypass the step where attention masks are applied, making them mask incompatible.

To address these challenges, we propose a novel approach aimed at significantly reducing the memory footprint, computational complexity, and energy consumption of ECCT, thereby enhancing its viability for real-world applications. Our method introduces three key innovations: (i) Weight quantization to the ternary domain through *Adaptive* Absolute Percentile (AAP) quantization. (ii) Head Partitioning Self Attention (HPSA), an efficient multi-head self-attention mechanism tailored for bipartite graph message passing (MP), designed to reduce computational complexity and runtime. (iii) Spectral positional encoding (SPE) of the Tanner graph by processing its Laplacian eigenspace. The Tanner graph Laplacian eigenspace forms a meaningful local coordinate system, providing structural information that is lost with ECCT's binary masking, without affecting inference runtime.

Our experimental results, conducted across a diverse range of codes, demonstrate that this approach not only matches, and in some cases exceeds, the performance of ECCT, but also offers computational complexity comparable to that of Belief Propagation (BP) Pearl (1988). These findings represent a significant step towards making transformer-based error correction practical for communication systems with limited computational resources, potentially bridging the gap between advanced neural decoding techniques and traditional efficient algorithms, such as BP.

## 2 RELATED WORK

Neural decoders for ECC have evolved from model-based methods, which implement parameterized versions of classical BP (Nachmani et al., 2016; 2018; Nachmani & Wolf, 2019; Caciularu et al., 2021), to model-free approaches utilizing general NN architectures (Kim et al., 2018; Gruber et al., 2017; Bennatan et al., 2018; Cammerer et al., 2017; Choukroun & Wolf, 2024a). A significant advancement in this field is the ECCT (Choukroun & Wolf, 2022b; 2024a;b), which, along with its extension using a denoising diffusion process (Choukroun & Wolf, 2022a), has achieved SOTA performance across various codes. These neural decoders primarily target short to moderate-length codes, addressing scenarios where classical decoders may not achieve optimal performance. Subsequently, Park et al. (2023; 2024) demonstrated improved performance, but at the expense of increased computational cost.

Transformers, while being powerful architectures, are resource-intensive. In response to the need to optimize large language models (LLMs), numerous quantization methods have been developed (Gholami et al., 2021; Wan et al., 2024; Zhu et al., 2024). These techniques fall into two categories: post-training quantization (PTQ) (Choukroun et al., 2019; Frantar et al., 2023; Chee et al., 2024) and quantization-aware training (QAT). Due to the resource-intensive nature of LLMs, recent studies have focused mainly on PTQ because of its low computational requirement and training overhead. However, PTQ often utilizes high-precision parameters, making it difficult to fully exploit the efficiency of quantization. In contrast, QAT has higher potential for accuracy but generally requires more resources and time, leaving research on QAT of LLMs in its preliminary stages (Jeon et al., 2024). Despite these challenges, notable work has emerged in QAT for LLMs. Wang et al. (2023) demonstrated effective quantization of weights to {-1, 1} values and activations to 8-bit integers.

An enhanced approach by Ma et al. (2024) introduced an additional zero weight value and utilized Abs-mean quantization, highlighting a correlation between model size and performance degradation post-quantization.

Recent efforts to address the computational limitations of self-attention mechanisms in transformers have focused on acceleration techniques. One approach approximates the self-attention function, reducing its computational cost from quadratic to linear time complexity (Choromanski et al., 2020). Other methods, such as those proposed by Child et al. (2019) and Beltagy et al. (2020), combine local and global attention to improve efficiency. Zaheer et al. (2020) further refines these methods by incorporating random global connections. The Reformer (Kitaev et al., 2020) explores Locality-Sensitive Hashing attention, while Shazeer (2019) introduces multi-query attention with shared keys and values across attention heads. Building on this, Ainslie et al. (2023) presents grouped-query attention, which uses fewer key-value heads to achieve results comparable to multi-head attention, but with faster computation. Additionally, Pope et al. (2023) introduces an optimized key-value cache mechanism to accelerate inference time.

Transformers have also been applied to graph-structured data, introducing graph structure as a soft inductive bias to address limitations of Graph neural networks (GNNs), such as over-squashing (Alon & Yahav, 2021; Topping et al., 2022). Dwivedi & Bresson (2020) proposed using Laplacian eigenvectors as PEs, while Kreuzer et al. (2021) incorporated Laplacian eigenvalues and used a dedicated Transformer for structural encoding. Building on these approaches, Rampášek et al. (2022) further improved performance by integrating innovations such as Signet (Lim et al., 2022), which addresses the sign ambiguity of eigenvectors, random-walk PE (Dwivedi et al., 2022), and PE based on the gradients of eigenvectors (Beaini et al., 2021).

## 3 SETTING AND BACKGROUND

**Problem Settings** We assume a standard transmission protocol that uses a linear code $C \subset \{0,1\}^n$. The code is defined by a binary generator matrix $G \in \{0,1\}^{k \times n}$ and a binary parity check matrix $H \in \{0,1\}^{(n-k) \times n}$, satisfying $GH^T = 0$ over $GF(2)$. The parity check matrix bipartite graph representation is referred to as the Tanner graph, which consists of $(n-k)$ check nodes and $n$ variable nodes. Linear codes encode information into structured codewords, enabling error detection and correction. The generator matrix $G$ maps messages to codewords, and the parity check matrix $H$ imposes constraints that define valid codewords. The transmission process begins with a $k$-bit input message $m \in \{0,1\}^k$, transformed into an $n$-bit codeword $x \in C$ via $G$, satisfying $Hx = 0$. This codeword is transmitted via a Binary-Input Symmetric-Output channel, resulting in a channel output $y = x_s + \epsilon$, where $x_s$ represents the Binary Phase Shift Keying modulation of $x$, and $\epsilon$ denotes random noise. The protocol ensures resilience against noise, allowing the decoder to recover the codeword from $y$. The decoding function $f : \mathbb{R}^n \to \mathbb{R}^n$ aims to provide a soft approximation $\hat{x} = f(y)$ of the original codeword. Following Bennatan et al. (2018); Choukroun & Wolf (2022b), a preprocessing step is applied to ensure codeword invariance and prevent overfitting present in model-free solutions. This yields a $(2n-k)$-dimensional vector $\tilde{y} = h(y) = [|y|, s(y)]$, where $|y|$ denotes $y$'s magnitude, and $s(y) \in \{0,1\}^{(n-k)}$ is the binary syndrome, computed as $s(y) = Hy_b := H\mathrm{bin}(y) := H(0.5(1 - \mathrm{sign}(y)))$. Preprocessing extracts the magnitude $|y|$ and syndrome $s(y)$, which summarize signal strength and error patterns. The codeword soft prediction takes the form $\hat{x} = y \odot \hat{\tilde{\epsilon}}$, where $\hat{\tilde{\epsilon}}$ denotes the prediction of *multiplicative* noise $\tilde{\epsilon}$ defined such that $y = x_s \odot \tilde{\epsilon} = 1 + \epsilon \odot x_s$. In our framework, the parameterized model is explicitly defined as $\tilde{x}_s = y \odot f_\theta(h(y))$, where $f_\theta$ represents our parameterized decoder.

**Error Correction Code Transformer (ECCT)** The ECCT (Choukroun & Wolf, 2022b) is a neural decoder based on the Transformer encoder architecture (Vaswani et al., 2017). Its input, $h(y) = [|y|, 1 - 2s(y)] \in \mathbb{R}^{2n-k}$, is embedded into a high-dimensional space, forming $\Phi \in \mathbb{R}^{(2n-k) \times d}$. The embedding matrix is processed by $N$ Transformer encoder blocks using Code-Aware Self-Attention (CASA):

$$A_H(Q, K, V) = \mathrm{Softmax}\left(d^{-\frac{1}{2}}(QK^T + g(H))\right)V,$$

where $g(H) : \{0,1\}^{(n-k) \times n} \to \{-\infty, 0\}^{(2n-k) \times (2n-k)}$ is a binary mask derived from $H$, removing connections between bits separated by more than two steps in the Tanner graph. The binary

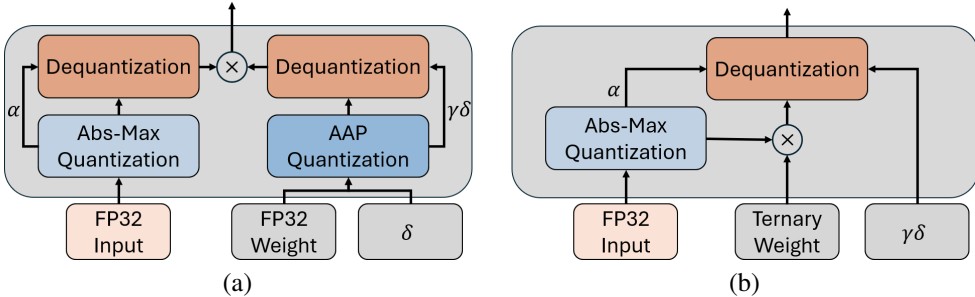

Figure 1: AAP Linear Layer: (a) QAT: Training with quantization noise; (b) Inference: Matrix multiplication using only integer additions with fixed ternary weights and fixed weight scale.

mask $g(H)$ ensures that self-attention focuses only on closely related bits, reflecting the structure of the code's Tanner graph. This improves the model's ability to capture local dependencies in the code. As the bit embeddings pass through each Transformer encoder block, they are iteratively refined by the self-attention mechanism, which dynamically emphasizes relationships between bits according to the structure imposed by $g(H)$. This allows the model to propagate and integrate local and global information about the code across multiple layers. The final block's output undergoes two projections to produce the noise prediction $\hat{\epsilon}$:

$$\hat{\epsilon} = W_o^T (W_{d \to 1} \Phi),$$

where $W_{d \to 1} \in \mathbb{R}^{d \times 1}$ reduces the embedding dimension and $W_o \in \mathbb{R}^{(2n-k) \times n}$ maps the result to the output space.

## 4 METHOD

Our proposed method enhances ECCT through several key modifications designed to improve both performance and efficiency. The primary enhancements are as follows:

1. We replace all linear layers within the Transformer blocks with our novel *Adaptive* Absolute Percentile (AAP) Linear layers. This modification introduces an adaptive quantization approach, achieving ternary weight representation and thereby improving the model's efficiency.

2. We introduce a novel self-attention mechanism, HPSA, which supersedes the CASA used in ECCT (Choukroun & Wolf, 2022b). HPSA significantly reduces memory footprint, computational complexity, and runtime, thus enhancing the overall efficiency of the model. To the best of our knowledge, our approach is the first to map the structure of the graph into patterns, with each group of heads within the multihead self-attention mechanism applying a specific pattern.

3. We incorporate the SPE derived from the Tanner graph's Laplacian eigenspace. This approach is inspired by Kreuzer et al. (2021)'s method of injecting a soft inductive bias of the graph's structure into the model, enabling the integration of a fine-grained connectivity absent in ECCT's binary mask.

4. To further optimize the model's efficiency, we replace (Mirzadeh et al., 2023) Gaussian Error Linear Units (GeLUs) (Hendrycks & Gimpel, 2016) with Rectified Linear Units (ReLUs).

5. We introduce a two-phased training process to enhance the model's performance.

This change simplifies the activation function to a thresholding operator which further contributes to complexity reduction.

### 4.1 ADAPTIVE ABSOLUTE PERCENTILE QUANTIZATION

Ternary quantization of a single precision tensor involves the element-wise assignment to one of three bins: {-1, 0, +1}. This results in $3^n$ possible arrangements for each weight tensor, where $n$

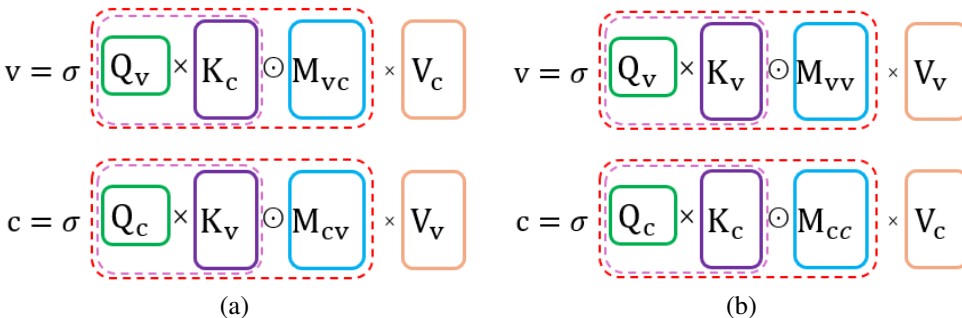

Figure 2: Head Partitioning Self-Attention: (a) First-ring and (b) second-ring head attention mechanisms. $Q$, $K$, $V$ denote query, key, and value tensors for variable (v) or check (c) nodes. $v =$ and $c =$ indicate new representations for variable and check nodes, respectively. $M_{cv}$, $M_{vc}$, $M_{cc}$, $M_{vv}$ are HPSA masks (see Fig. 3). $\sigma$ denotes the Softmax function.

is the tensor's number of elements. In NNs with numerous weights, finding the optimal arrangement becomes infeasible due to this highly exponential number of options. Existing approaches, such as abs-mean quantization (Ma et al., 2024) often struggle to achieve the right sparsity for precise management of feature retention and elimination, making certain desirable weight distributions extremely difficult to attain during training.

To address this challenge, we propose a novel method that provides maximum flexibility to the model. Our *Adaptive* Absolute Percentile (AAP) quantization method aims to identify the appropriate percentile of absolute values to use as a scaling factor. This percentile is optimized during training, thereby defining the desired sparsity and structure at the finest granularity. For each weight tensor (excluding biases) during each training forward pass, we calculate the $p$-th percentile, for a predefined $p$, of the absolute values of the weights, denoting this value as $\gamma$. The value of $\gamma$ depends solely on the current weight distribution and changes with each training iteration. The scale is then computed as $\gamma\delta$, where $\delta$ is a learnable parameter initialized to one. This approach allows $\delta$ to adjust the percentile dynamically throughout training, helping the model effectively balance sparsity and information retention for each weight matrix.

In contrast to existing methods, which either rely on a weight distribution-based scale (e.g., Ma et al. (2024)) or use a learnable scale that may be initialized with a calibration set (e.g., Jeon et al. (2024)), we combine both approaches. The computed scale $\gamma\delta$ is then used to scale the entire weight matrix. Finally, each scaled weight is rounded to the nearest integer among $\{-1, 0, +1\}$.

$$\text{Ternary}(W) = \text{RoundClip}\left(\frac{W}{\gamma \cdot \delta + \varepsilon}, -1, 1\right)$$
$$\text{RoundClip}(x, a, b) = \max(a, \min(b, \text{round}(x)))$$
$$\gamma = \text{Percentile}(\text{Abs}(W), 0.5)$$

(1)

where $\text{Percentile}(x, p)$ returns the $p$-th percentile value of $x$, and $\text{Abs}(x)$ computes the element-wise absolute values of $x$. The activations undergo Absmax quantization to INT8 as follows:

$$\text{Quant}(x) = \text{RoundClip}\left(x \times \frac{Q_b}{\alpha}, -Q_b, Q_b\right)$$

(2)

where $\alpha = \|x\|_\infty$ and $Q_b$ is the maximum value for the INT8 quantization range. Similarly to Wang et al. (2023); Ma et al. (2024), $\alpha$ is not fixed during inference. The complete quantization scheme, incorporating both weight and input quantization, operates as follows:

$$\text{AAPLinear}(x, W, b) = \text{Quant}(x) \cdot \text{Ternary}(W) \cdot \frac{\gamma\delta\alpha}{Q_b} + b$$

(3)

Here, $x \in \mathbb{R}^{m \times n}$ is the FP32 layer's input, $W \in \mathbb{R}^{n \times p}$ is the FP32 weight, and $b \in \mathbb{R}^p$ is the FP32 bias. The product of the quantized weights and input is dequantized before bias addition. All scaling

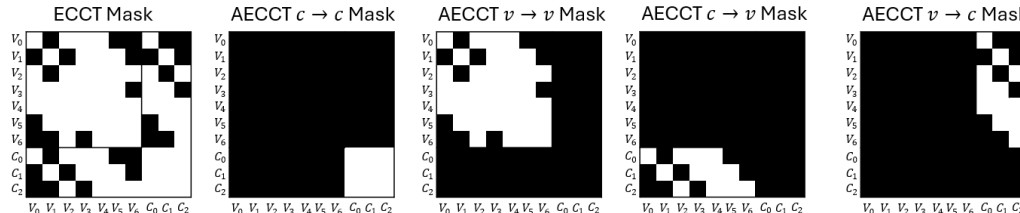

Figure 3: Code-aware masks of Hamming(4,7). AECCT masks utilize two distinct patterns, with each head applying only one: either first-ring or second-ring MP. First-ring MP uses c-to-v and v-to-c masks, while second-ring MP employs v-to-v and c-to-c masks. In contrast, the ECCT mask (on the left) applies both first and second rings for all heads. AECCT masks exhibit greater sparsity compared to ECCT, leading to reduced computational complexity.

factors, $\delta$, $\gamma$, and $\alpha$, are scalars, which enhances computational efficiency. Figure 1 illustrates the AAP mechanism during both the training and inference phases. The method avoids floating-point matrix multiplication, relying primarily on integer addition and subtraction operations, significantly reducing computational complexity.

## 4.2 HEAD PARTITIONING SELF ATTENTION

While the CASA mechanism of ECCT has demonstrated effective performance in decoding, we aim to further optimize its computational efficiency since we seek to develop neural decoders with complexity comparable to their classical counterparts such as BP. To this end, we introduce Head Partitioning Self Attention (HPSA), which maintains the effectiveness of CASA while significantly reducing computational complexity. HPSA strategically divides ECCT's masking via the attention heads into two groups: first-ring and second-ring MP heads. This division not only enhances efficiency but also introduces a graph-structure inductive bias by distinguishing between neighbors and second-ring connections, in contrast to the Code-Aware mask in ECCT. An illustration of HPSA is provided in Figure 2.

**First Group: First-Ring Message Passing** This group of heads performs attention between nearest neighbors in the Tanner graph. This process, which we term first-ring MP, facilitates communication between variable nodes and check nodes. The corresponding attention masks are the $c \rightarrow v$ and $v \rightarrow c$ in Figure 3, demonstrating the increased sparsity of HPSA compared to the Code-Aware mask from ECCT.

**Second Group: Second-Ring Message Passing** The second group focuses on what we call second-ring connections. These heads apply attention only between nodes at a distance of two in the Tanner graph. This allows for MP between variable nodes and other variable nodes, as well as between check nodes and other check nodes. The corresponding attention masks are the $c \rightarrow c$ and $v \rightarrow v$ in Figure 3, further illustrating the sparsity enhancement of HPSA.

By structuring the attention mechanism, HPSA achieves results comparable to CASA while drastically reducing complexity. This approach brings the computational efficiency of our method closer to that of the BP algorithm, moving us significantly closer to practical implementation in resource-constrained environments.

## 4.3 POSITIONAL ENCODING OF THE TANNER GRAPH

Although the two-rings connectivity code-aware mask has proven effective in ECCT, it provides the model with limited information about the Tanner graph's structure. By design, it does not distinguish between first-ring and second-ring connections (Choukroun & Wolf, 2024a). To enhance the decoder's performance beyond this limitation, we propose incorporating a soft inductive bias through SPE induced by the Tanner graph. This approach, inspired by Kreuzer et al. (2021), injects information from the Laplacian eigenspace, which serves as a meaningful local coordinate system, thereby enriching the model's understanding of the graph's topology. Intuitively, the Lapla-

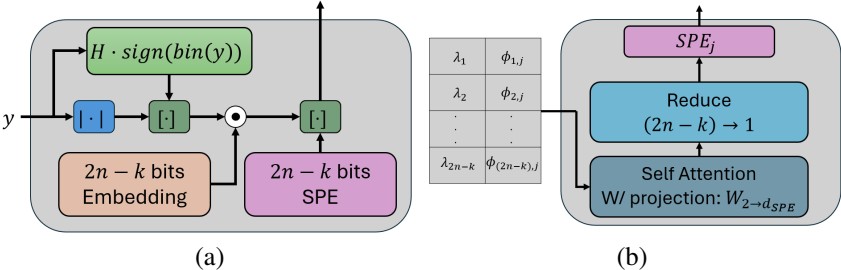

(a)                    (b)

Figure 4: Tanner PE. (a) The SPE matrix is concatenated to the initial nodes' embedding matrix. (b) Creation of the SPE vector for individual node $j$, which is then concatenated with the node's embedding. $\lambda_i$ denotes the i-th smallest eigenvalue of the Tanner graph. $\phi_i$ denotes the eigenvector corresponding to the i-th smallest eigenvalue and $\phi_{i,j}$ is its j-th element.

cian eigenspace provides a way to encode the relationships between nodes in the graph, allowing the model to "see" the graph's structure beyond simple connectivity. This enables the decoder to better understand the role of each node in the overall topology. The following procedure is applied for each node $j$ in the Tanner graph, as illustrated in Figure 4:

$$\text{SPE}_j = W_{(2n-k)\to 1}\text{MHSA}(Q_j, K_j, V_j) \tag{4}$$

$$Q_j = K_j = V_j = W_{2\to d_{\text{PE}}}\Phi_j \tag{5}$$

where $\Phi_j \in \mathbb{R}^{(2n-k)\times 2}$ is constructed by concatenating the graph's eigenvalues with the $j$-th node's corresponding values in the eigenvectors, $d_{\text{SPE}}$ is a hyperparameter, $W_{2\to d_{\text{SPE}}}$ is a learnable tensor, $W_{(2n-k)\to 1}$ is a reduction operator (e.g., linear projection, max/average pooling), and MHSA denotes Multi-Head Self-Attention. The resulting vector $\text{SPE}_j$ serves as the PE for node $j$ and is concatenated with the node's embedding. This process is repeated for all nodes in the graph. At inference, the learned SPE vectors remain fixed, removing the extra runtime computation present during training.

## 5 ANALYSIS

**Compression Rate** The linear layers in the ECCT model constitute over 95% of the total weight count, including the channel's output embedding. By employing ternary values, which theoretically require only 1.58 bits for representation, we achieve significant compression. Replacing FP32 values with ternary values results in a 95% reduction in the memory footprint of these layers. Consequently, the AECCT's overall memory footprint is reduced to approximately 10% of the original ECCT, achieving a compression rate of around 90%.

**Energy Consumption** Energy consumption is a critical factor, especially when deploying the AECCT on edge devices or in data centers, as it directly impacts battery life and operational costs. We base our analysis on energy consumption models for addition and multiplication operations on 7nm and 45nm chips for FP32 and INT8, as outlined by Horowitz (2014); Zhang et al. (2022); Wang et al. (2023). Our findings indicate that the AECCT achieves substantial energy savings. Specifically, it reduces the energy consumption of arithmetic operations in linear layers by at least 224 times on 7nm chips and 139 times on 45nm chips, compared to the original ECCT.

**Complexity** Dedicated hardware optimized for this approach avoids attention calculations that are subsequently masked out by the code-aware mask. Assuming such hardware, for a Tanner graph $T = (V, E)$, the ECCT CASA's complexity in a single Transformer encoder block is $\mathcal{O}(d(\sum_{x_i\in V} d_{x_i} + \beta_{x_i}))$, where $d$ is the embedding vector size, $d_{x_i}$ is the degree of vertex $x_i$, and $\beta_{x_i}$ denotes the number of vertices with a distance of two from $x_i$, derived from applying both first- and second-ring MP in each head. In contrast, our HPSA approach reduces this complexity to $\mathcal{O}(d_h h_f(\sum_{x_i\in V} d_{x_i}) + d_h h_s(\sum_{x_i\in V} \beta_{x_i}))$, where $h_f$ and $h_s$ correspond to the number of first-ring and second-ring heads, respectively, and $d_h$ is the head dimension. This reduction stems from the partitioning of attention heads, where each head is dedicated exclusively to either first-ring or

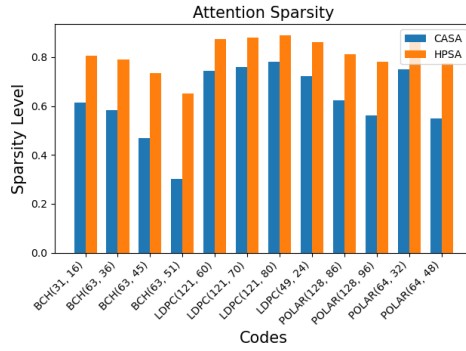
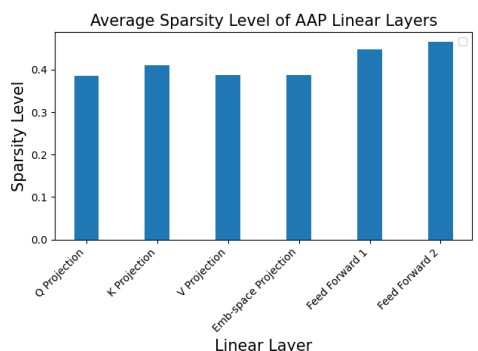

Figure 5: Comparison of attention sparsity levels for HPSA with $h_f = h_s = 4$. Sparsity level represents the proportion of query-key dot products avoided relative to a full pairwise attention mechanism.

Figure 6: AAP weight sparsity levels for models trained on BCH(63,51) codes. Each bar represents the average sparsity across all instances of a specific linear layer type in the Transformer encoder blocks.

second-ring MP. Figure 5 illustrates this complexity reduction by comparing the number of query-key dot products avoided in CASA and HPSA for various codes. The sparsity level is defined as the percentage of dot products avoided relative to quadratic pairwise attention. As shown, HPSA achieves sparsity levels of at least 78% across most codes, while the CASA's sparsity ranges from 30% to 78%. This visual representation corroborates our theoretical analysis, demonstrating the significant computational efficiency gained through HPSA. By strategically partitioning attention heads and dedicating them to specific ring levels, HPSA dramatically reduces the number of necessary dot product calculations, resulting in a more efficient and optimized attention mechanism.

The AECCT model's complexity is governed by parameters $N$, $d$, $h_f$, and $h_s$, offering exceptional flexibility in balancing accuracy and computational efficiency. As demonstrated by Choukroun & Wolf (2022b), even the most modest ECCT architectures (e.g., $N = 2$, $d = 32$) consistently outperform BP across several codes. This performance advantage extends to AECCT, which not only maintains this superior decoding capability but does so with complexity comparable to BP. As illustrated in Figure 7, the shallowest AECCT architecture, with complexity comparable to BP, outperforms BP with 50 iterations (L=50). This showcases AECCT's ability to offer superior performance even at its most basic configuration, achieving a balance between computational efficiency and decoding capability that ECCT could not attain. The performance gap widens as we increase $N$ and $d$ in AECCT, since increasing the number of BP iterations beyond 50 yields only marginal improvements. Further analysis of AECCT's complexity is provided in Appendix A.

**AAP Sparsity** We analyzed the sparsity level of the Adaptive Absolute Percentile (AAP) linear layers in the AECCT for a model trained on BCH(63,51) code. Figure 6 illustrates our findings, revealing that the percentage of zero-valued weights ranges from approximately 40% to 50%. Importantly, this sparsity effectively reduces the dimension of the embedding vectors to around $0.45d$, further amplifying the efficiency gains discussed in our complexity analysis.

# 6 EXPERIMENTS[1]

**Training & Inference** We utilize a Post-Layer Normalization (Post-LN) Transformer architecture, consistent with the original Transformer design in Vaswani et al. (2017), distinguishing it from the ECCT approach, as we empirically found it to be more effective. The training process is divided into two phases.

In the first phase, we train the ECCT from scratch, incorporating several modifications: ReLU activations replace GeLUs, HPSA is used instead of CASA, and learnable Tanner graph PE is integrated.

---

[1]Code is available at `https://github.com/aecct-paper/AECCT`

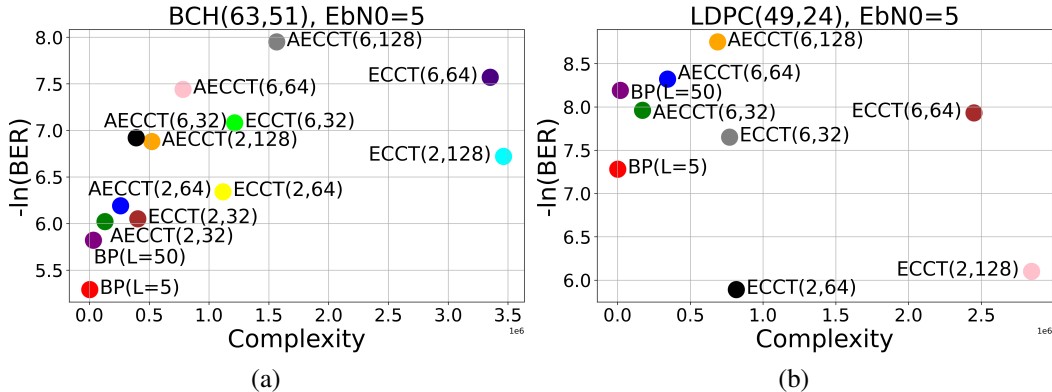

Figure 7: Accuracy vs. Complexity analysis for (a) BCH(63,51), (b) LDPC(49,24), at $E_b/N_0 = 5$. We compare AECCT with BP and ECCT in terms of decoding quality versus complexity. AECCT($N$, $d$) denotes an architecture with $N$ Transformer encoder blocks with embedding size $d$, with a similar notation used for ECCT. BP($L$) denotes BP with $L$ iterations.

The model undergoes training for 1000 epochs (1500 for $N = 10$), with each epoch consisting of 1000 batches. We employ the Adam optimizer with a batch size of 128.

In the second phase, the linear layers within the encoder blocks are replaced with AAP-linear layers, initialized using the weights obtained from the first phase. QAT is then applied using the same configuration as in the first phase. Upon completion, the weights of the AAP-linear layers are fixed as ternary values, and their corresponding scales are also fixed.

Throughout both phases, following the approach of Choukroun & Wolf (2022b), we use a zero codeword with a Gaussian channel sampled from a normalized SNR ($E_b/N_0$) range of 3 to 7. The learning rate is initialized at $10^{-4}$ and decays to $5 \times 10^{-7}$ following a cosine schedule. The cross-entropy loss is used to guide the model in learning the multiplicative noise (Bennatan et al., 2018).

**Results** We evaluate our proposed method on three types of linear block codes: Low-Density Parity Check (LDPC) codes (Gallager, 1962), Polar codes (Arikan, 2009), and Bose–Chaudhuri–Hocquenghem (BCH) codes (Bose & Ray-Chaudhuri, 1960), using parity check matrices from Helmling et al. (2019). The architecture is defined by two key parameters: the number of encoder layers ($N$) and the embedding dimension ($d$). Performance is assessed by measuring bit error rates (BER) across a range of $E_b/N_0$ values, followed by Choukroun & Wolf (2022b). Table 1 presents our results, showing the negative natural logarithm of the BER. We compare the performance of our AECCT to the ECCT (with Pre-LN architecture) and BP (Pearl, 1988) across two different architectures: $N = 6$ and $N = 10$, both with an embedding dimension of $d = 128$. The ECCT architecture employed in our experiments is based on the implementation described in Choukroun & Wolf (2022b) and detailed in Section 3.. The results indicate that the AECCT performs on par with the ECCT, and in some cases, even exceeds it for certain codes, while remaining much more efficient.

Figure 8 shows BER and BLER comparisons for the POLAR(64,48) code between AECCT, ECCT, and SCL (Tal & Vardy, 2012), with SCL results presented for a list length of $L = 1$. The SCL experiments are conducted by us, using the code framework of (Cassagne et al., 2019). Additional BER and BLER curves can be found in Appendix E.

**Ablation Study** Our comprehensive ablation study evaluates the key components of our proposed model, with results detailed in Table 2. We use an ECCT model with Post-Ln architecture as our baseline, then separately incorporate each AECCT component to assess its individual impact. First, we examine the **impact of HPSA**. The results demonstrate that HPSA maintains or improves performance relative to the CASA-based baseline, while simultaneously reducing computational complexity. This dual benefit of preserved or enhanced effectiveness coupled with increased efficiency underscores HPSA's value as a key component of our model. Next, we investigate the **influence**

Table 1: We present the performance of our proposed method against established baselines, measured using -log(BER) across three normalized SNR levels. The negative logarithm transformation of BER is employed for clearer visualization, with larger values representing superior error correction capabilities. We compare our AECCT to the ECCT as our baseline. Additionally, we compare it to BP Pearl (1988) with $L = 5$ (first row) iterations and $L = 50$ (second row) iterations. We separate the comparison between ECCT and AECCT according to the number of encoder blocks, $N$. For each $N \in \{6, 10\}$, bold text indicates the best results between ECCT and AECCT for that specific $N$ value. Notably, for BCH codes, $N = 10$ for AECCT was unnecessary, as AECCT with $N = 6$ outperforms ECCT with $N = 10$.

| Method | BP | | | ECCT $N = 6$ | | | AECCT $N = 6$ | | | ECCT $N = 10$ | | | AECCT $N = 10$ | | |
|---|---|---|---|---|---|---|---|---|---|---|---|---|---|---|---|
| | 4 | 5 | 6 | 4 | 5 | 6 | 4 | 5 | 6 | 4 | 5 | 6 | 4 | 5 | 6 |
| Polar(64,48) | 3.52 4.04 4.48
4.26 5.38 6.50 | | | 6.36 | 8.46 | 11.09 | **6.43** | **8.54** | **11.12** | 6.43 | 8.40 | 11.00 | **6.54** | **8.51** | **11.12** |
| Polar(128,86) | 3.80 4.19 4.62
4.49 5.65 6.97 | | | **6.31** | **9.01** | **12.45** | 6.04 | 8.56 | 11.81 | 7.26 | 10.60 | **14.80** | **7.28** | 10.60 | 14.59 |
| Polar(128,96) | 3.99 4.41 4.78
4.61 5.79 7.08 | | | **6.31** | **9.12** | **12.47** | 6.11 | 8.81 | 12.15 | **6.85** | **9.78** | 12.90 | 6.79 | 9.68 | **12.93** |
| LDPC(49,24) | 5.30 7.28 9.88
6.23 8.19 11.72 | | | 5.79 | 8.13 | 11.40 | **6.10** | **8.65** | **12.34** | 6.35 | 9.01 | 12.43 | **6.67** | **9.35** | **13.56** |
| LDPC(121,60) | 4.82 7.21 10.87
- - - | | | 5.01 | 7.99 | 12.78 | **5.17** | **8.32** | **13.40** | 5.51 | 8.89 | 14.51 | **5.71** | **9.31** | **14.90** |
| LDPC(121,70) | 5.88 8.76 13.04
- - - | | | 6.19 | 9.89 | 15.58 | **6.38** | **10.1** | **16.01** | 6.86 | 11.02 | 16.85 | **7.05** | **11.40** | **17.30** |
| LDPC(121,80) | 6.66 9.82 13.98
- - - | | | 7.07 | 10.96 | 16.25 | **7.27** | **11.50** | **16.90** | 7.76 | 12.30 | 17.82 | **7.98** | **12.60** | **18.10** |
| BCH(31,16) | 4.63 5.88 7.60
- - - | | | 6.39 | 8.29 | 10.66 | **7.01** | **9.33** | **12.27** | 6.41 | 8.30 | 10.77 | **7.21** | **9.47** | **12.45** |
| BCH(63,36) | 3.72 4.65 5.66
4.03 5.42 7.26 | | | 4.68 | 6.65 | 9.10 | **5.19** | **6.95** | **9.33** | **5.09** | **6.96** | **9.43** | 4.90 | 6.64 | 9.19 |
| BCH(63,45) | 4.08 4.96 6.07
4.36 5.55 7.26 | | | 5.60 | 7.79 | 10.93 | **5.90** | **8.24** | **11.46** | 5.72 | 7.99 | 11.21 | **5.83** | **8.15** | **11.52** |
| BCH(63,51) | 4.34 5.29 6.35
4.50 5.82 7.42 | | | 5.66 | 7.89 | 11.01 | **5.72** | **8.01** | **11.24** | 5.38 | 7.40 | 10.50 | **5.68** | **7.88** | **11.04** |

Table 2: Evaluation of AECCT components against (Post-Ln) ECCT. For added generality, we used $(N = 6, d = 64)$ for LDPC(49,24) while maintaining $(6, 128)$ for other codes as in Tab. 1.

| Model | POLAR(64,48) | BCH(31,16) | LDPC(49,24) |
|---|---|---|---|
| ECCT | 6.40 8.50 11.10 | 6.95 9.21 12.04 | 5.97 8.44 12.01 |
| ECCT + HPSA | 6.40 8.52 11.17 | 7.00 9.24 12.12 | 5.98 8.48 12.12 |
| ECCT + SPE | 6.43 8.53 11.10 | 7.01 9.21 12.31 | 5.97 8.46 12.09 |
| ECCT + HPSA + SPE | 6.50 8.61 11.15 | 7.00 9.25 12.07 | 6.01 8.48 12.01 |
| ECCT + AP | 6.39 8.49 11.14 | 7.05 9.27 12.33 | 5.90 8.34 11.70 |
| ECCT + Abs-Mean | 6.40 8.49 11.07 | 7.02 9.22 12.25 | 5.86 8.29 11.50 |
| ECCT + AAP | 6.41 8.51 11.13 | 7.06 9.37 12.37 | 5.91 8.35 11.74 |
| **AECCT**: HPSA + SPE + AAP | 6.43 8.54 11.12 | 7.01 9.33 12.27 | 5.89 8.33 11.67 |

**of the SPE**. We find that integrating positional and structural information from the Tanner graph's Laplacian through SPE significantly boosts overall model performance. To ensure a fair comparison, we maintain consistent total embedding dimensions by reducing the size of the channel's output embedding vectors before concatenating the SPE vectors. Finally, we assess the **impact of AAP quantization**. Our analysis shows that AAP quantization outperforms absolute percentile (AP) quantization. The adaptive approach introduces a learnable parameter $\delta$, enabling dynamic adjustment of weight sparsity and effectively controlling feature filtration. Additionally, we compared AAP quantization to ECCT with abs-mean quantization (Ma et al., 2024), achieving superior

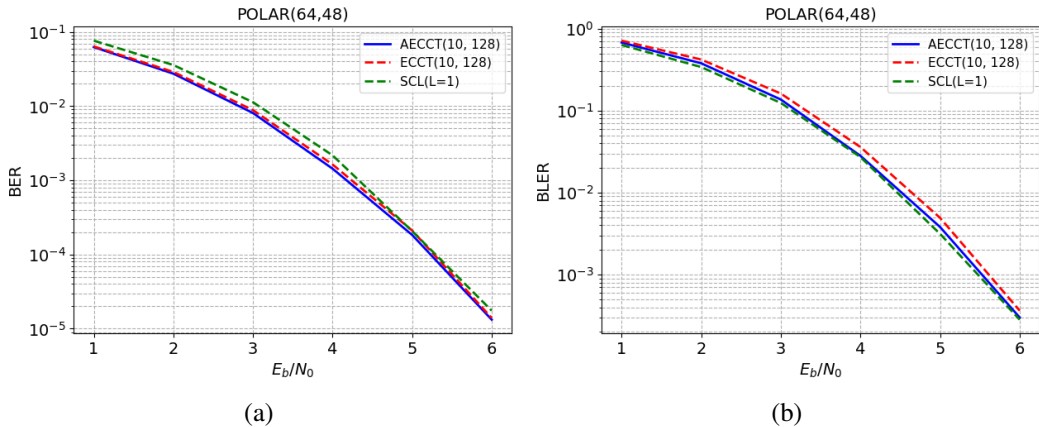

Figure 8: (a) BER; (b) BLER comparisons of AECCT, ECCT, and SCL for the POLAR(64,48) code. The architectures for AECCT and ECCT are configured with $N = 10$ encoder blocks and an embedding dimension of $d = 128$.

results on every code tested. This demonstrates that AAP quantization surpasses the current SOTA in ternary quantization.

Appendices B and C present additional ablation studies. The former evaluates AECCT with varying numbers of first and second ring heads, revealing their similar importance with optimal performance when $h_f = h_s$. The latter compares a binary weighted version of AECCT to (our) ternary weighted AECCT, both using AAP quantization. Results demonstrate the superiority of ternary representation, achieving substantial performance gains with minimal bit usage increase (1.58 vs 1), justifying our choice of ternary quantization. We analyze in Appendix D the necessity of $\delta$ in the AAP method, demonstrating its importance for dynamic thresholding across different AAP layers.

## 7 CONCLUSIONS

We introduced the AECCT, an enhanced version of the ECCT initially proposed by Choukroun & Wolf (2022b). The AECCT integrates several novel techniques: Adaptive Absolute Percentile Quantization, which compresses the linear layer weights in the Transformer encoder blocks to ternary values; Head Partitioning Self-Attention, which replaces the code-aware self-attention module, significantly reducing complexity; and Tanner Graph Positional Encoding, which improves the model's overall effectiveness. The AECCT achieves a complexity level comparable to BP while reducing memory usage, energy consumption, and computational complexity, all while delivering performance on par with the ECCT. Altogether, these enhancements bring transformer-based error correction decoders closer to practical deployment in real-world communication systems, offering notable improvements in the reliability of physical layer communications. As future work, we wish to explore learned Tanner-graph-based positioning techniques and apply pattern-based head partitioning to other structured learning problems. In addition, we wish to explore implementing dedicated hardware that can leverage the ternary-weight linear layer, which requires only integer additions and subtractions, and efficiently support sparse self-attention mechanism to maximize the computational benefits of the proposed method.

In a broader context, our work addresses a gap in the literature regarding the acceleration of small Transformers, particularly those where attention patterns are dictated by the problem domain. The novel quantization method we propose enables exact localized adaptations, and the head partitioning method we propose addresses any hierarchical or structured data.

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

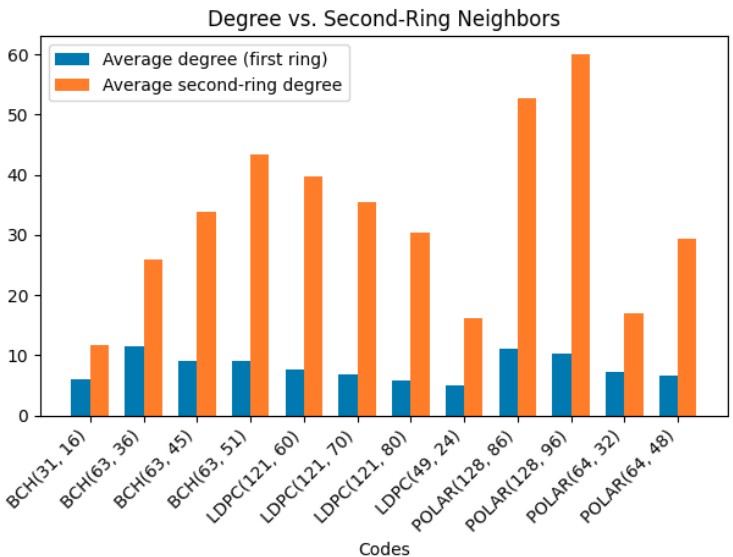

Figure 9: The expected values of $\beta_{x_i}$, representing the number of vertices two edges away from $x_i$, are compared to the expected values of $d_{x_i}$, the degree of $x_i$, for $d = 32$ and $d_h = 4$. The size of $\beta_{x_i}$ significantly affects the complexity of second-ring heads in the HPSA.

## A  COMPLEXITY ANALYSIS

In this section, we provide a detailed breakdown of the complexity for various components of our AECCT model, focusing on the AAP linear layer, the Head Partitioning Self-Attention (HPSA) mechanism, and the second-ring degree $\beta$.

**AAP Linear Complexity**   We analyze the complexity of the AAP linear layer by separating it into multiplication and addition components. The complexity for FP32 multiplications, which arises from the quantization of the input activation matrix and the dequantization of the output activation matrix, is given by

$$2(2n - k)d = 2|V|d = \mathcal{O}(|V|d), \tag{6}$$

where $T = (V, E)$ is the Tanner graph, $d$ is the embedding vector size, $k$ denotes the input message size, and $n$ is the output vector size of the channel. Matrix multiplication, which involves only additions and subtractions, results in an INT8 addition complexity of

$$(2n - k)d^2 = \mathcal{O}(|V|d^2). \tag{7}$$

The bias addition, performed in FP32, is $\mathcal{O}(|V|d)$.

**HPSA Complexity**   Similarly, we decompose the complexity of HPSA into multiplications and additions. Assuming an equal number of first- and second-ring heads, the total number of FP32 multiplications for all first-ring heads in a single Transformer encoder block is

$$\left( \sum_{i=1}^{n} d_i + \sum_{i=1}^{n-k} \tilde{d}_i \right) \frac{d}{2} = 2|E|\frac{d}{2} = \mathcal{O}(|E|d), \tag{8}$$

where $d_i$ denotes the degree of the $i$-th variable node and $\tilde{d}_i$ denotes the degree of the $i$-th parity check node. The number of FP32 additions is similar.

The total number of FP32 multiplications required for all second-ring heads in a single Transformer encoder block is

$$\frac{d}{2} \sum_{x_i \in V} \beta_{x_i}, \tag{9}$$

where $\beta_{x_i}$ represents the number of vertices at a distance of two edges from $x_i$. Again, the number of FP32 additions is similar.

Table 3: Complexity comparison between AECCT and (sum-product) BP, assuming for simplicity that the number of first-ring heads equals the number of second-ring heads in the HPSA. We analyze the magnitude of $\beta$ for several codes in A.1. The AECCT involves FP32 multiplications, FP32 additions, and INT8 additions. Therefore, we analyze each of these operations separately.

| Operation | AECCT | BP |
|---|---|---|
| FP32 MUL | $\mathcal{O}(Nd(\|V\| + \|E\| + \sum_{x_i \in V} \frac{\beta_{x_i}}{2}))$ | $\mathcal{O}(L\|E\|)$ |
| FP32 ADD | $\mathcal{O}(Nd(\|E\| + \sum_{x_i \in V} \frac{\beta_{x_i}}{2}))$ | $\mathcal{O}(L\|E\|)$ |
| INT8 ADD | $\mathcal{O}(Nd^2\|V\|)$ | - |

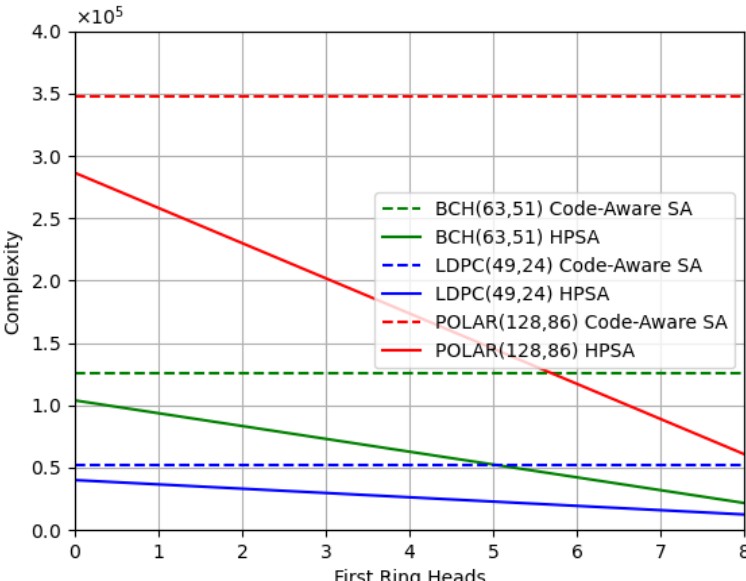

Figure 10: The impact of $h_f$ and $h_s$ on HPSA's complexity is analyzed for three codes: BCH(63,45), LDPC(49,24), and POLAR(128,86). We calculate the number of multiplications required for the CASA and compare it to HPSA with all possible combinations of $h_f$ and $h_s$, where $h_f$ represents the number of first-ring heads and $h_s$ the number of second-ring heads.

**AECCT Complexity** Having examined the complexities of individual components, we now combine these to determine the total complexity of AECCT. The results of this combined analysis are presented in Table 3.

A.1 IMPACT OF $\beta$

We analyze the expected value of $\beta_{x_i}$ to evaluate its influence on computational complexity, as illustrated in Figure 9. Our analysis indicates that $\mathbb{E}[\beta_{x_i}]$ is approximately $\frac{\mathbb{E}[d_{x_i}]^2}{2}$. Given that second-ring heads exhibit higher complexity, it is feasible to employ more first-ring heads. Figure 10 demonstrates the complexity of the HPSA for various combinations of $h_f$ and $h_s$, compared to the CASA mechanism. The results clearly show that HPSA significantly reduces complexity compared to CASA.

A.2 COMPLEXITY VS BER

We analyze the trade-off between complexity and performance for AECCT, ECCT, and BP in Figure 11. The results are presented for $E_b/N_0$ of 4 and 6. Our findings indicate that AECCT is compa-

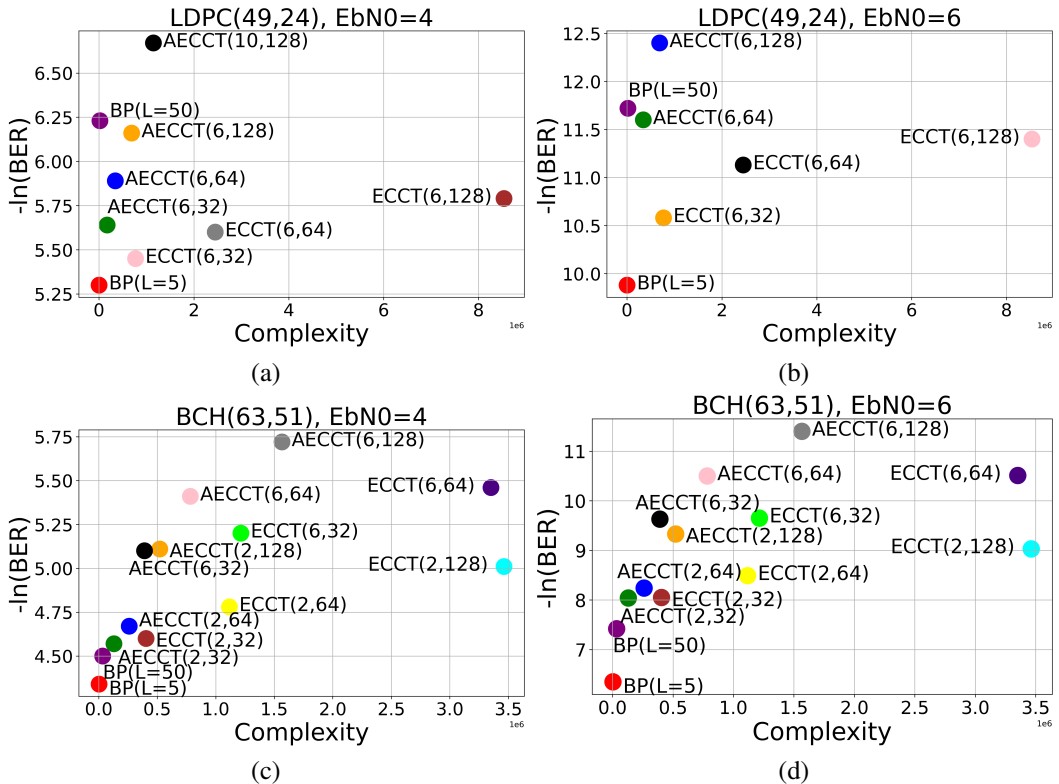

Figure 11: -ln(BER) vs. Complexity for LDPC(49,24) at $E_b/N_0$ values of (a) 4 and (b) 6, and BCH(63,51) at $E_b/N_0$ values of (c) 4 dB and (d) 6 dB. We compare AECCT, BP, and ECCT in terms of performance and complexity. AECCT($N$, $d$) denotes an architecture with $N$ Transformer encoder blocks and an embedding size of $d$, with a similar notation used for ECCT. BP($L$) refers to BP with $L$ iterations.

Table 4: Ablation of $h_f$ and $h_s$

| Code | $N$, $d$ | 1st ring, 2nd ring | Neg ln(BER) |
|------|----------|--------------------|-------------|
| BCH(31,16) | 6, 128 | 2, 6 | 6.91 9.24 12.1 |
| BCH(31,16) | 6, 128 | 4, 4 | 7.00 9.25 12.1 |
| BCH(31,16) | 6, 128 | 6, 2 | 6.95 9.25 12.1 |

rable to BP in both complexity and performance, while demonstrating better scalability. Moreover, AECCT architectures consistently outperform ECCT architectures with significantly lower complexity. For example, AECCT with $N = 6$ and $d = 64$ consistently surpasses ECCT with $N = 6$ and $d = 32$, while also being more computationally efficient.

# B  FIRST & SECOND RING HEADS BALANCE

We analyze the optimal configuration of first-ring and second-ring heads in the HPSA mechanism. We evaluate AECCT without the AAP contribution, varying the number of first-ring heads $h_f \in \{2, 4, 6\}$ and setting $h_s = 8 - h_f$, where $h_s$ represents the number of second-ring heads. Table 4 presents these findings, indicating that both types of heads contribute similarly, with $h_f = h_s = 4$ yielding the best results.

Table 5: AECCT binary vs AECCT ternary

| Code | $N, d$ | AECCT ternary | AECCT binary |
|---|---|---|---|
| LDPC(49,24) | 6, 128 | 6.10 8.65 12.3 | 5.96 8.42 11.9 |
| BCH(31,16) | 6, 128 | 7.01 9.33 12.1 | 6.52 8.55 11.0 |
| POLAR(64,48) | 6, 128 | 6.37 8.52 11.1 | 6.12 8.20 10.6 |

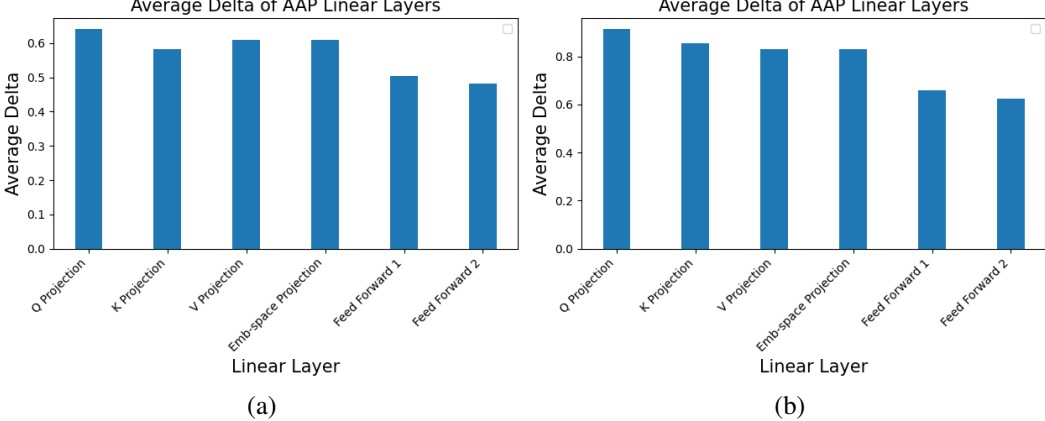

(a)                                           (b)

Figure 12: Analysis of $\delta$. The average $\delta$ value of each type of linear layer across the AECCT trained on (a) BCH(63,51); (b) LDPC(121,70).

## C  TERNARY VS BINARY PRECISION CHOICE

We evaluate our AECCT method using binary precision with AAP quantization as an alternative to the ternary precision AAP quantization. The results are listed in Table 5. Evidently, the ternary quantization significantly outperforms the binary one in terms of precision. This substantial performance improvement, achieved with only a minimal increase in bit usage (1.58 vs 1), strongly supports our decision to use ternary over binary quantization.

## D  AAP DYNAMIC FEATURE CONTROL

We present the post-training values of $\delta$ for two AECCT models in Figure 12. Notably, the $\delta$ values are higher for the self-attention projections, leading to a greater elimination of features, whereas in the feed-forward layers, $\delta$ retains more information. This behavior may be explained by the fact that self-attention (through the query, key, and value projections) focuses on a specific subset of features for each attention head, while the feed-forward layers primarily reduce redundancy between blocks without drastically limiting the feature set.

## E  ADDITIONAL RESULTS

We present BER and BLER curves for three codes: POLAR(128,86), BCH(63,45), and BCH(63,51). For the POLAR(128,86) code, comparisons include AECCT, ECCT, and SCL decoding, with SCL results shown for a list length of $L = 1$. For the BCH codes, BCH(63,45) and BCH(63,51), the curves compare the performance of AECCT and ECCT, demonstrating the effectiveness of AECCT. These results further validate the improvements introduced by AECCT in terms of decoding accuracy (see Figure 13).

## F  GENERALIZATION OF AAP QUANTIZATION TO OTHER DOMAINS

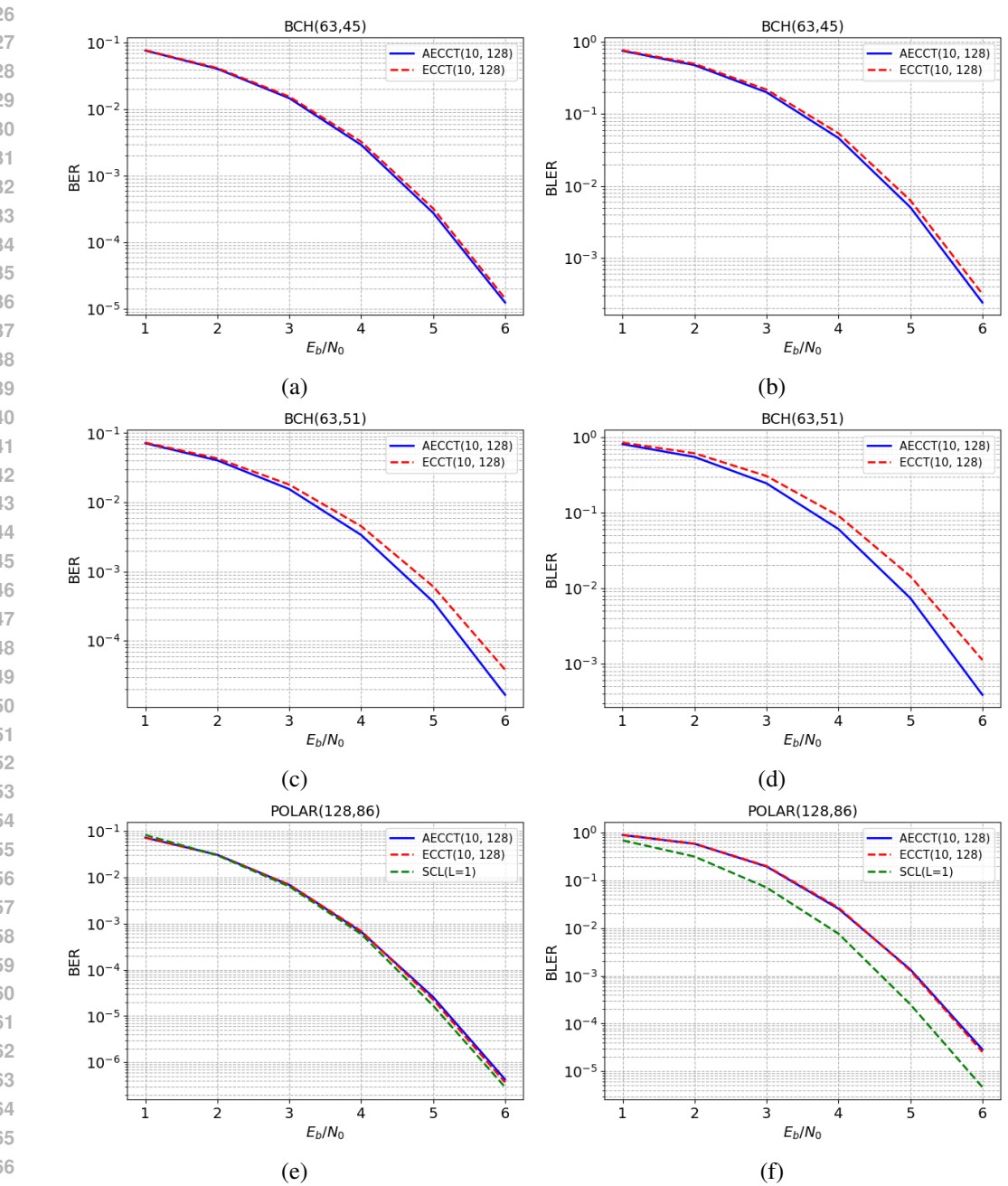

Figure 13: BER and BLER curves for additional codes. Subfigures (a) and (b) present BER and BLER results for BCH(63,45), respectively, while subfigures (c) and (d) show BER and BLER results for BCH(63,51). Subfigures (e) and (f) present BER and BLER results for POLAR(128,86), including comparisons between AECCT, ECCT, and SCL decoding (Tal & Vardy, 2012), with SCL results shown for a list length of $L = 1$. These results demonstrate the performance of AECCT across different codes and validate its effectiveness.

To evaluate the generalization of AAP quantization beyond error correction, we conducted an experiment using TinyBERT (Jiao et al., 2020) on the SST-2 sentiment classification dataset (Socher et al., 2013). Starting with a pretrained TinyBERT model, we fine-tuned it for one epoch and subsequently applied Quantization-Aware Training for one additional epoch using two quantization methods: AAP and abs-mean quantization. As shown in Table 6, AAP quantization outperformed

abs-mean, a method considered state-of-the-art for ternary quantization, by achieving higher accuracy. This result highlights AAP's effectiveness in generalizing to other domains, showcasing its potential to perform well on diverse tasks.

Table 6: Accuracy on SST-2 dataset using TinyBERT.

| Method | Accuracy (%) |
|---|---|
| FP32 (Full Precision) | 88.7 |
| AAP Quantization | 83.0 |
| Abs-mean Quantization | 82.6 |

