# OpenReview forum: "Accelerating Error Correction Code Transformers"
_ICLR.cc/2025/Conference — Submitted to ICLR 2025_

### Official Review · Reviewer_mcns · 2024-10-31

**Soundness:** 3
**Presentation:** 2
**Contribution:** 2
**Rating:** 3
**Confidence:** 5

**Summary:**

This paper presents a method to enhance the efficiency and practicality of transformer-based error correction code (ECC) decoders, specifically targeting the error correction code transformer (ECCT) model introduced in prior work. The authors propose several modifications to ECCTs, including ternary weight quantization, a novel attention mechanism called head partitioning self-attention (HPSA), and a spectral positional encoding inspired by the Tanner graph's eigenspace.
Jointly, these improvements lead to reduce memory, energy, and computational requirements while maintaining or improving the performance of the original ECCTs.

**Strengths:**

The paper proposes interesting methods like HPSA and adaptive absolute percentile quantization, which could be valuable outside of coding applications.

**Weaknesses:**

While the proposed approach introduces certain novel elements, I believe the paper falls short in critical areas, particularly regarding contribution scope and experimental results. Therefore, I recommend rejection for the following reasons:

1. Limited contribution: The paper proposes interesting methods like HPSA and adaptive absolute percentile quantization, which could be valuable outside of coding applications. However, since this paper is fundamentally a paper in coding theory, these contributions remain peripheral to the main focus (furthermore, per se they are not enough to warrant publication at ICLR).
And within the realm of coding theory, the contribution of the paper is limited.

While relevant for the coding community, the contribution lacks the depth and broader impact expected at a major machine learning conference or in a top coding journal. A more appropriate fit would be a coding conference such as ISIT or ITW.

Furthermore, although ML-based decoding of conventional codes is a growing research area, with already many published papers, this paper does not provide a method that competes with conventional state-of-the-art codes/decoders.


2. Weak comparisons and benchmarking: The experimental results are insufficient to substantiate the relevance of the proposed decoder.

Specifically, the authors compare the performance of their transformer-based decoder to that of belief propagation decoding for various codes, including polar codes and BCH codes. BP decoding of BCH codes and polar codes is known to be highly suboptimal. There are well-known, more efficient decoding algorithms for these codes (note that the considered BCH codes are very short). Hence, the current results provide no hint on whether the proposed decoders are good or not.

To establish relevance, the paper would need to demonstrate comparable or superior performance to state-of-the-art decoding techniques in terms of accuracy or complexity. However, even if such comparisons were provided, in my opinion the contribution is still insufficient for a high-profile venue.

3. Use of non-standard metrics: The paper presents performance using the negative natural logarithm of the BER. Is there any reason for not adhering to standard metrics such as bit error rate curves or block error rate (BLER) curves (the latter being preferable)?
The use of this unconventional (to day the least) metric complicates unnecessarily the interpretation of the results. I do not think it is wise to come up with an awkward way of showing results, when there is a established way to present results for error correcting codes.

Furthermore, reporting performance for only a few specific values of $E_b/N_0$ provides an incomplete view; it's unclear whether phenomena like error floors might occur at higher values.

Including conventional BLER curves over a range of $E_b/N_0$ values would make the results easier to interpret and align with community standards. Without such curves, it's challenging to assess the effectiveness of the proposed decoder and, in my opinion, the paper should be accepted in any venue.

**Questions:**

1. As mentioned above, the authors should report BLER curves.

2. The authors provide results for a number of LDPC codes. However, there are no details regarding these codes. Are them the best-available LDPC codes in the literature, or some designed by the authors?

3. Related to Comment 2 and my comments in "Weaknesses," the authors should rework their results section and compare the performance of the proposed approach with that of state-of-the-art codes/decoders

---

> ### Author Response · Authors · 2024-11-18
> **Official Comment by Authors - First part**
>
> Thank you for your thoughtful feedback.
>
> > However, since this paper is fundamentally a paper in coding theory, these contributions remain peripheral to the main focus (furthermore, per se they are not enough to warrant publication at ICLR). And within the realm of coding theory, the contribution of the paper is limited.
>
> The paper is not a coding theory paper and should not be judged as such. It belongs to a substantial and growing body of work on neural decoders, which is published in the ML community. This paper addresses a critical goal: bringing neural decoders closer to real-world applicability. The required tools for this task are developed within the deep learning community. The proposed methods, including HPSA and adaptive absolute percentile quantization, are novel and generic machine learning tools with broad applicability beyond coding theory.
>
> > While the proposed approach introduces certain novel elements, I believe the paper falls short in critical areas, particularly regarding contribution scope and experimental results.
>
> To address concerns about experimental results, we have added BER and BLER curves for a more comprehensive evaluation. Our contributions encompass both acceleration and error correction, introducing spectral positional encoding in this context for the first time and demonstrating that our quantization method outperforms state-of-the-art ternary quantization techniques. The results for the ablation study models are presented below:
>
> | Code         | Abs-Mean https://arxiv.org/abs/2402.17764 | AAP                  |
> |--------------|---------------------------------------------|----------------------|
> | POLAR(64,48) | 6.40  8.49  11.07                          | 6.41  8.51  11.13   |
> | LDPC(49,24)  | 5.86  8.29  11.50                          | 5.91  8.35  11.74   |
> | BCH(31,16)   | 7.02  9.22  12.25                          | 7.06  9.37  12.37   |
>
> We also show that AAP can generalize to other domains. We took the TinyBERT model https://arxiv.org/abs/1909.10351, a 14M-parameter model well-suited for evaluating small-model quantization, and used the SST-2 dataset as in its original paper. The results are summarized below:
>
> | Model                                   | Abs-Mean | AAP  |
> |-----------------------------------------|----------|------|
> | huawei-noah/TinyBERT_General_4L_312D    | 82.6     | 83.0 |
>
> Given that the FP32 accuracy of TinyBERT in our experiments was 88.7%, these results clearly demonstrate that AAP outperforms Abs-Mean, the SOTA method for ternary quantization, in terms of accuracy and applicability to small transformers.
>
> > Weak comparisons and benchmarking: The experimental results are insufficient to substantiate the relevance of the proposed decoder.
>
> Thank you for raising this concern. First and foremost, our focus is on reaching and surpassing ECCT's performance while ensuring the model is significantly faster and more lightweight. We successfully achieved this, demonstrating that our approach matches or exceeds ECCT's effectiveness while drastically reducing runtime and memory requirements.
>
> We appreciate your feedback and have addressed it by adding additional results comparing our method to SCL for polar codes, which we believe provide stronger evidence of the decoder's relevance. While SCL decoding is specifically optimized for Polar codes, AECCT demonstrates comparable performance with low L values. This highlights the versatility of AECCT, as it can be trained to decode any code, yet still achieves results on par with an algorithm tailored specifically for Polar codes.
>
> | Code         | SCL \( L=1 \) | SCL \( L=4 \) | AECCT (\( N=10, d=128 \)) |
> |--------------|---------------|---------------|---------------------------|
> | POLAR(64,48) | 6.13  8.30  11.02 | 6.63  8.70  11.13 | 6.54  8.51  11.12     |
> | POLAR(128,86)| 7.61  10.96  15.00 | 9.30  12.92  17.30 | 7.28  10.60  14.59   |
> | POLAR(128,96)| 6.83  9.46  13.20 | 8.15  11.65  18.29 | 6.79  9.68  12.93     |
>
> > To establish relevance, the paper would need to demonstrate comparable or superior performance to state-of-the-art decoding techniques in terms of accuracy or complexity. However, even if such comparisons were provided, in my opinion, the contribution is still insufficient for a high-profile venue.
>
> ECCT is regarded as the state-of-the-art among neural decoders for many families of codes. Additionally, we have included a comparison to SCL in the paper, presented through BER and BLER curves. A detailed complexity analysis is provided in Appendix A to address this concern.

---

> > ### Comment · Reviewer_mcns · 2024-11-23
> >
> > Thank you very much for your reply. I appreciate the effort you have taken to address my concerns and provide additional results. However, your response has further reinforced my opinion of the paper. Below, I provide additional comments and clarifications:
> >
> > 1. Is this a coding paper or not?
> >
> > This is unequivocally a coding paper. In your response, you state: "It belongs to a substantial and growing body of work on neural decoders." Note that, neural decoders are inherently methods for decoding error-correcting codes and therefore belong within the domain of coding (theory). Consequently, it is critical to evaluate the proposed neural decoders against the state-of-the-art in coding and decoding techniques as a whole--not just against other neural decoders. This leads directly to my second point.
> >
> > 2. Weak comparisons and benchmarking
> >
> > In my original review, I noted: "The experimental results are insufficient to substantiate the relevance of the proposed decoder." While I appreciate that you have now included BLER plots for additional comparisons, these results unfortunately confirm my initial concern: the proposed neural decoder performs significantly worse than the state-of-the-art.
> >
> > Take, for example, your new Figure 13.f. Your decoder performs worse than the polar code decoded with successive cancellation (SC) with a list size of 1. More importantly, this comparison is not fair. A proper comparison should be made against a polar code with CRC and a larger list size, as used in current communication standards. Under such conditions, at a BLER of $10^-4$, the proposed decoder would perform more than 1.5 dB worse--a significant gap that highlights its limitations.
> >
> > 2.a Choice of Code Parameters
> >
> > To facilitate fair and meaningful comparisons, I strongly recommend evaluating performance on standard code parameters widely used in the field. For example, the (128,64) code is commonly benchmarked in the literature (see, e.g., Liva et al., "Code Design for Short Blocks: A Survey"). Comparisons using these customary parameters would provide clearer insights into the relevance and competitiveness of your method.
> >
> > In any case, clearly your method performs poorly.
> >
> > 2.b Presentation of Results
> >
> > The current comparisons may mislead reviewers and readers unfamiliar with coding theory. No expert in coding would consider a polar code with SC and list size 1 to be a valid benchmark for modern decoding performance. While this may be unintentional, it creates the impression that the authors are aware of the method's shortcomings relative to state-of-the-art decoders and are avoiding direct comparisons. Such selective benchmarking undermines the credibility of the work.
> >
> >
> > Conclusion
> >
> > In light of these points, I must reaffirm my evaluation of the paper. I cannot raise my score and strongly believe that this paper should be rejected.

---

> > > ### Author Response · Authors · 2024-11-23
> > >
> > > We appreciate the reviewer’s response. We appear to diverge on the "meta" question of the overarching goals of contemporary research in the field of neural decoding, which has led to a difference in perspectives.
> > >
> > > > …it is critical to evaluate the proposed neural decoders against the state-of-the-art in coding and decoding techniques as a whole--not just against other neural decoders. This leads directly to my second point.
> > >
> > > Given the numerous published works on neural decoding, our work focuses on accelerating the state-of-the-art ECCT neural decoder, and we successfully deliver on this task.
> > >
> > > The reviewer’s claim can be equally applied to every published work on neural decoding from major ML conferences, as ECCT is the SOTA neural decoder. The field of neural decoding has been gaining considerable traction, including a recent publication in [Nature](https://www.nature.com/articles/s41586-024-08148-8), which employs a Transformer-based decoder and references a follow-up work of ECCT in the field of quantum error correction.
> > >
> > > By adopting the narrow perspective of comparing universal neural decoders with specific families of codes and decoders developed over decades, the review overlooks the machine learning contributions that advance representation learning for decoding.
> > >
> > > > While I appreciate that you have now included BLER plots for additional comparisons, these results unfortunately confirm my initial concern.
> > >
> > > This claim ignores the fact that our model is trained to optimize the BER objective, similar to most neural decoders. While there are contributions that perform (non-differentiable) optimization over BLER, this is not the focus of our work.
> > >
> > > Polar codes have been extensively studied over an extended period, with the SCL decoder emerging as a well-established and highly optimized decoding method. It demonstrates exceptional effectiveness and performance specifically for polar codes. In contrast, transformer-based decoders are still in the early stages of exploration and are not exclusively targeted at polar codes. While their performance shows great promise, there is considerable room for improvement in both decoding efficiency and complexity, underscoring the need for research such as ours.
> > >
> > > Traditional decoders are typically code-specific, designed to work exclusively with particular types of codes. For example, the BP decoder is well-suited for LDPC codes, the BM decoder for BCH codes, and the SCL decoder for polar codes. A notable advantage of the published ECCT and its extensions is their versatility and universality, demonstrating effective performance across a wide range of code classes.
> > >
> > > > Choice of Code Parameters/ Presentation of Results
> > >
> > > Just as it would not be fair—or even possible—to expect the SCL decoder to outperform LDPC codes or vice versa, neural decoders cannot, **for now**, outperform all specialized decoders for every SOTA code.
> > >
> > > However, this limitation should not hinder the development of the neural ECC field. For instance, recent research has demonstrated that codes optimized with respect to a Transformer-based neural decoder’s inductive bias can be trained to produce optimal neural codes [Choukroun & Wolf, 2024c].
> > >
> > > Our method is not intended to compete on accuracy but to accelerate the ECCT network, which has proven its utility in several recent advancements in the field. Moreover, it serves as a significant case study for accelerating small networks, a topic that has been largely overlooked.
> > >
> > > Nonetheless, we will provide the performance metrics for both the baseline and our method on the specified POLAR code. Additionally, we will include results for SCL decoding with longer list sizes. We note that a same-term comparison should include a CRC-aided neural decoder, which is beyond the scope of our current work. This is because, once again, our goal is not to compete on accuracy or to provide code or standard-specific solutions.

---

> ### Author Response · Authors · 2024-11-18
> **Official Comment by Authors - Second part**
>
> > Use of non-standard metrics: The paper presents performance using the negative natural logarithm of the BER. Is there any reason for not adhering to standard metrics such as bit error rate curves or block error rate (BLER) curves (the latter being preferable)?
>
> Thank you for your comment. We have added BER and BLER curves to the paper as requested. The use of the negative natural logarithm of the BER aligns with a line of previous work in the field.
>
> > The authors provide results for a number of LDPC codes. However, there are no details regarding these codes. Are them the best-available LDPC codes in the literature, or some designed by the authors?
>
> Thank you for pointing this out. All the codes used were sourced from https://rptu.de/channel-codes and represent various families of sparse codes.
>
> > Related to Comment 2 and my comments in "Weaknesses," the authors should rework their results section and compare the performance of the proposed approach with that of state-of-the-art codes/decoders.
>
> Answered above.

---

> > ### Comment · Reviewer_mcns · 2024-11-23
> >
> > Thank you for providing the BLER results. These results are indeed helpful for evaluating the performance of the proposed decoder. However, they confirm my initial concern: the proposed decoder performs significantly worse than the state-of-the-art.
> >
> > Regarding the LDPC codes, it appears that the codes considered in your experiments are not the best-performing ones available in the literature.

---

### Official Review · Reviewer_3y8a · 2024-11-01

**Soundness:** 3
**Presentation:** 4
**Contribution:** 3
**Rating:** 8
**Confidence:** 3

**Summary:**

The paper has three main thrusts that I can see, and it targets these towards specifically, acceleration and compression of small transformer model architectures used for error-correction code decoding in communications systems. This is an incremental but significant upgrade to earlier work on ECC transformer decoders. The main thrusts are:
1) Quantization: The paper uses an adaptive layer-based quantization scheme that combines maximum-based quantization with distribution-based quantization schemes to provide a ternary quantization for the network, greatly accelerating its computation (claimed) while also compressing model size.
2) Head Partitioning: Masking attention heads based on the Tanner graph node connections at varying levels of relation (first- and second-ring masks) allows greater sparsity in attention computation while also preserving performance.
3) Positional Encoding: A soft positional encoding is also introduced as an inductive bias in the tokenizer/learnable encoder (from what I can see) that further preserves signal positional information and compensates for information loss in the attention head masking.

**Strengths:**

1) Solid presentation, all major thrusts are clearly presented and detailed. The paper also provides diagrams and examples to ease understanding and reading in what can be presented as a very dense, mathematical topic that a reviewer would have trouble grasping or reading through.
2) Analysis and experimentation include results on hardware for energy consumption reduction, a key metric in model compression. It also includes ablation studies for different encoding methods and acceleration due to model compression. All in all it analyzes most of the ablations relevant to the topic. I particularly commend the hardware results for different transistor nodes.
3) Contributions while building clearly off of a prior design and prior work, are cleverly tuned to the problem domain and a very clear niche. In particular the use of the Tanner graph and selective masking shows adaptation of the transformer compression problem to the given domain.

**Weaknesses:**

1) Concluding statements mention that the paper shows a general approach for the compression problem in small transformers - I would say that this is a bit of a stretch - the Tanner subgraph masking and the adjustments to positional encoding are not applicable to all problem domains, leaving the model quantization formulation the only really general cross-domain contribution. That formulation has not been benchmarking in the experimental section against prior art, and its acceleration alone (as opposed to the entire framework) has not been tested in the main document. I would say that ablation results for each component of the framework should be added in order to make this general claim.

2) A key contribution is replacing GeLU with ReLU in the bullet points of the front matter. This is absolutely not novel, as has been examined in [1] just to name a single source. I would strongly advise omitting this claim or qualifying it to the given problem domain.
[1] https://arxiv.org/abs/2310.04564 [Addressed]

3) While this work builds on the prior ECCT work, I would want a presentation of the model architecture in the main draft and clearly marked so that the scale of the 'small transformer' is visible, rather than having to dig through the ECCT paper and assume that this is the paper used. [Addressed]

4) The work assumes dedicated hardware for the transformer decoder in Section 5 when calculating complexity. This may not be a realistic assumption in practice due to the specialized and expensive design process for accelerator ASICs. I would want to see complexity calculations on a common edge accelerator platform such as an NVIDIA Hopper architecture or a Coral edge TPU, or a citation for the architecture-level simulator presumably used to generate these numbers (e.g. ScaleSim). Numbers for hardware without that backing are unreliable.

5)  The sparsity induced by masking is structured, and therefore may be applicable to standard accelerators. Why is there no mention of this? I do not see an analysis of the benefits of the structured sparsity from the masks in Section 4.2 compared to unstructured sparsity, which is a major plus point that has not been exploited.

**Questions:**

1) See point 5 in Weaknesses - is the sparsity structured, and if so, does your assumed hardware architecture exploit that?

2) See (4) in Weaknesses - Where is this assumption coming from and what edge platform architecture was used? Is there a citation or a cycle-accurate simulation of the architecture? Analytical complexity numbers that assume optimal hardware are unrealistic.

3) What are the authors' thoughts on the contribution of each component of this framework to the overall acceleration? Is there an ablation study that can be presented in the main doc? If not, a small table and rough numbers would be good so that we can see how much general approaches like quantization contribute and how much the domain-specific tweaks such as Tanner graph masking contribute.

See discussion - the authors have conditionally addressed much of the questions above and paper weaknesses. I am editing the review to reflect this.

---

> ### Author Response · Authors · 2024-11-18
> **Official Comment by Authors - First part**
>
> Thank you for your thoughtful feedback and for raising these points.
>
> > Concluding statements mention that the paper shows a general approach for the compression problem in small transformers - I would say that this is a bit of a stretch.
>
> As highlighted in the paper, HPSA is a versatile method that can be applied across many domains to achieve both acceleration and compression of self-attention mechanisms. This makes it broadly applicable to data-structured problems in machine learning. For example, in multi-modal problems, a subgroup of self-attention heads can be allocated to each modality, showcasing its adaptability to various challenges.
>
> Furthermore, we emphasize that our Adaptive Absolute Percentile (AAP) quantization method represents a significant contribution to the compression of small Transformers. To address your concern about benchmarking, we conducted additional experiments comparing AAP with Abs-Mean quantization by Ma et al. (2024) https://arxiv.org/abs/2402.17764, recognized as the state-of-the-art method for ternary quantization. These results demonstrate AAP's advantages over Abs-Mean, establishing its effectiveness as a general-purpose quantization method for small Transformers.
>
> The experiments included the three models from the ablation study (POLAR(64,48), BCH(31,16), and LDPC(49,24)) and the TinyBERT model https://arxiv.org/abs/1909.10351, a 14M-parameter model well-suited for evaluating small-model quantization. For TinyBERT, we used the SST-2 dataset as in its original paper. The results are summarized below:
>
> ### Results for Ablation Study Models:
> | Code        | Abs-Mean          | AAP              |
> |-------------|-------------------|------------------|
> | POLAR(64,48)| 6.40  8.49  11.07 | 6.41  8.51  11.13 |
> | LDPC(49,24) | 5.86  8.29  11.50 | 5.91  8.35  11.74 |
> | BCH(31,16)  | 7.02  9.22  12.25 | 7.06  9.37  12.37 |
>
> ### Results for TinyBERT:
> | Model                      | Abs-Mean | AAP  |
> |----------------------------|----------|------|
> | TinyBERT_General_4L_312D   | 82.6     | 83.0 |
>
> Given that the FP32 accuracy of TinyBERT in our experiments was 88.7%, these results clearly demonstrate that AAP outperforms Abs-Mean, the SOTA method for ternary quantization, in terms of accuracy and applicability to small Transformers.
>
> > I would strongly advise omitting the ReLU to GeLU claim, or qualify it against prior art.
>
> We have followed the reviewer's advice and included a citation to https://arxiv.org/abs/2310.04564.
>
> > While this work builds on the prior ECCT work, I would want a presentation of the model architecture in the main draft and clearly marked so that the scale of the 'small transformer' is visible, rather than having to dig through the ECCT paper and assume that this is the paper used.
>
> We revised the ECCT section to improve clarity, following the reviewer’s request. To illustrate how small ECCT is, consider the following: (1) it uses small embedding sizes (32 to 128), (2) it employs a limited number of encoder blocks (2 to 10), (3) the input sequences, representing codewords, are relatively short, with a maximum length of ~160—much smaller than those in LLMs, and (4) the final two linear projections are minimal, containing only ~20k parameters. Altogether, these factors result in a model with at most ~2M parameters.
>
> > See (4) in Weaknesses - Where is this assumption coming from and what edge platform architecture was used? Is there a citation or a cycle-accurate simulation of the architecture? Analytical complexity numbers that assume optimal hardware are unrealistic.
>
> Regarding complexity calculations on common edge accelerator platforms, it is important to note that ECC is not designed to run efficiently on GPUs or TPUs, as this would be extremely wasteful. Obtaining accurate complexity measurements for AECCT requires implementing dedicated hardware to fully leverage the AAP linear layers.
>
> To utilize the ternary-weight linear layer, we attempted to use the BitBLAS library https://github.com/microsoft/BitBLAS, which implements a 2-bit kernel, as demonstrated by Ma et al. (2024) https://arxiv.org/abs/2402.17764. Unfortunately, no improvement was observed due to the small sequence size. This issue is documented https://github.com/microsoft/BitBLAS/issues/118 , as the overhead of activation dequantization and quantization becomes significant when both the input and output features are small.

---

> > ### Comment · Reviewer_3y8a · 2024-11-23
> > **Addressed Comments**
> >
> > 1) Thank you for highlighting the broader applications of the method. Provided that this summary or a similar note on the wider applicability of HPSA is acknowledged in the paper and it is noted that this is specifically broadly applied for self-attention acceleration, I can mark that comment as addressed.
> > 2) This addressed my comments on GeLU to ReLU and the ECCT architecture summary more than adequately.
> > 3) Provided that the limited applicability to existing ASIC platforms such as GPUs and TPUs is noted in the paper and the use of BitBLAS is also noted (for reproducibility or wider research's sake), I would mark this comment (on hardware complexity calculations) likewise resolved.
> > I can therefore say that most of my comments have been conditionally resolved, provided some notes are provided in the main document or referenced in appendices.

---

> ### Author Response · Authors · 2024-11-18
> **Official Comment by Authors - Second part**
>
> > See point 5 in Weaknesses - is the sparsity structured, and if so, does your assumed hardware architecture exploit that?
>
> We appreciate the reviewer’s insightful comment regarding the structure of sparsity. The sparsity in both AECCT and ECCT is not structured. While we attempted to permute the sequence to create clusters of unmasked entries https://arxiv.org/abs/2001.04451, this proved to be a highly non-trivial task within the context of Tanner graphs.
>
> > Can the authors include a clearly marked discussion of the ECCT architecture used in Experiments? I would prefer to have that marked off and visible.
>
> Thank you for the suggestion. We have added a clearly marked discussion on the ECCT architecture used in the experiments, as recommended.
>
> > What are the authors' thoughts on the contribution of each component of this framework to the overall acceleration? Is there an ablation study that can be presented in the main doc? If not, a small table and rough numbers would be good so that we can see how much general approaches like quantization contribute and how much the domain-specific tweaks such as Tanner graph masking contribute.
>
> Thank you for your comment. A complexity analysis detailing the contributions of different components, assuming dedicated hardware, is provided in Appendix A.

---

### Official Review · Reviewer_L3we · 2024-11-02

**Soundness:** 3
**Presentation:** 3
**Contribution:** 3
**Rating:** 8
**Confidence:** 2

**Summary:**

This paper introduces a novel transformer based neural error correction architecture. It builds on the idea of the Error Correction Code Transformer (ECCT) by performing ternary quantization of the parameters in linear layers, by modifying the masking mechanism in the attention operation and also by introducing positional embeddings. The architecture is empirically validated against the original ECCT and the more traditional Belief Propagation algorithm and is shown to outperform both in terms of accuracy.

**Strengths:**

Through it's experimental results, the paper convincingly demonstrates the benefit of ternary weight quantization, Spectral Position Embedding and tailoring the attention mask for bipartite graph message passing over the previous state of the art transformer based method - the error correction code transformer (ECCT).

**Weaknesses:**

As an outsider to the field of error correction, I found that the description of the problem domain was a bit too brief. The paper would benefit from expanding the problem setting section. It would also benefit the paragraph describing the ECCT architecture if the equations weren't inlined and that the sequence of transformations defining the architecture were serialized.

**Questions:**

Merely out of curiosity: have the authors considered non transformer based architectures for this task? Presumably denoising diffusion models might have a role to play given the nature of the problem?

---

> ### Author Response · Authors · 2024-11-18
> **Official Comment by Authors**
>
> We sincerely thank the reviewer for their thoughtful and constructive comments, which have helped us improve the clarity and presentation of our paper.
>
> > As an outsider to the field of error correction, I found that the description of the problem domain was a bit too brief. The paper would benefit from expanding the problem setting section. It would also benefit the paragraph describing the ECCT architecture if the equations weren't inlined and that the sequence of transformations defining the architecture were serialized.
>
> We have expanded the Problem Setting section to provide a more accessible description of the problem domain. Additionally, we have revised the paragraph describing the Error Correction Code Transformer architecture by serializing the sequence of transformations and presenting the equations in a non-inline format, as suggested.
>
> > Merely out of curiosity: have the authors considered non-transformer based architectures for this task? Presumably denoising diffusion models might have a role to play given the nature of the problem?
>
> We appreciate your interest in exploring alternative architectures for this task. Indeed, we are actively considering applying the AECCT method to a new approach called DDECC https://arxiv.org/abs/2209.13533, which builds on the ECCT by incorporating denoising diffusion models. We believe this expansion has the potential to further enhance the decoding process and address challenges posed by noise in communication channels.
>
> Thank you again for your valuable input, and we hope the revisions align with your expectations.

---

> > ### Comment · Reviewer_L3we · 2024-11-26
> >
> > I thank the authors for their response, and very much appreciate the changes they made to their draft. My concerns are now adequately addressed.

---

### Official Review · Reviewer_mhUK · 2024-11-03

**Soundness:** 2
**Presentation:** 2
**Contribution:** 2
**Rating:** 5
**Confidence:** 4

**Summary:**

This paper addresses the challenges of high computational complexity and memory requirements faced by Error Correction Code Transformers (ECCT) in practical applications by introducing an innovative acceleration method. The approach centers on three technical innovations:
- a specially designed ternary weight quantization method (AAP) that enables multiplication-free linear layers.
- an optimized self-attention mechanism (HPSA) based on code-aware multi-head processing that significantly reduces computational complexity.
- a positional encoding scheme (SPE) through Tanner graph eigendecomposition that provides richer graph connectivity representation without affecting inference runtime.

Experimental results demonstrate that this method not only matches or surpasses the original ECCT's performance but also achieves a 90% compression ratio while reducing arithmetic operation energy consumption by at least 224 times on modern hardware, making transformer-based error correction more practical in resource-constrained environments.

**Strengths:**

This paper presents contributions to enhancing ECCT's practical applicability through a comprehensive optimization approach. The key strengths lie in its multi-faceted architectural improvements and efficient implementation strategies.

1. The paper introduces three well-designed technical innovations that directly address ECCT's structural limitations. The Head Partitioning Self Attention (HPSA) mechanism optimizes the self-attention computation specifically for bipartite graph message passing, while the Spectral Positional Encoding (SPE) leverages Tanner graph eigendecomposition to provide richer structural information without runtime overhead. These architectural improvements demonstrate a deep understanding of ECCT's underlying structure and successfully enhance its performance.

2. Except for two structural improvements of ECCT, the paper makes a step further to explore the possibility of low-bit quantization of ECCT. The paper uses a ternary weight quantization method (AAP) specifically designed for ECCT's compact architecture, achieving multiplication-free linear layers while maintaining model accuracy.

The comprehensive experimental validation demonstrates that these improvements maintain ECCT's decoding performance while reducing its computational complexity to a level comparable with traditional Belief Propagation methods. This work contributes to making transformer-based error correction more practical in resource-constrained environments and helps narrow the efficiency gap between neural decoding techniques and traditional algorithms.

**Weaknesses:**

The weaknesses of this work are manifested in the following three aspects:

1. Limited Scope and Generalization.
The method's exclusive focus on ECCT architecture raises concerns about its broader impact. As ECCT has not gained significant attention in academia nor found applications in industry, optimizations specific to this architecture may have limited practical value.

2. Lack of Technical Novelty in Quantization.
The proposed AAP quantization method, while showing good performance, primarily combines existing techniques - weight distribution analysis and learnable quantization steps. The approach does not present substantial innovation in quantization methodology, making it more of an implementation enhancement than a theoretical advancement.

3. Insufficient Validation of Acceleration Claims.
Despite positioning itself as an acceleration method, the paper's experimental validation concentrates on compression ratios, sparsity metrics, and energy consumption reduction. While these metrics demonstrate improved efficiency, the absence of direct inference speed measurements and real-world acceleration benchmarks leaves the actual acceleration effects unverified. This gap between claimed acceleration and demonstrated results weakens the paper's central premise.

Suggestions:

1. Enhanced Validation of Quantization Method

The paper would benefit from more comprehensive experiments comparing the proposed AAP method with existing quantization approaches. Specifically, comparative studies with methods like GPTQ and OmniQuant that focus solely on weight distribution or learnable scales would better demonstrate the advantages of combining these techniques. Such experiments would strengthen the paper's claims about AAP's effectiveness and innovation in quantization methodology.

2. Demonstration of Actual Acceleration Effects

The paper needs to provide concrete evidence of inference speedup to support its acceleration claims. We suggest adding experiments that measure and analyze the actual inference acceleration achieved after implementing all proposed optimizations. If practical acceleration is currently limited by implementation constraints, this limitation should be explicitly acknowledged, and the path toward achieving actual speedup in future work should be discussed. This transparency would better align the paper's claims with its demonstrated results.

**Questions:**

1.In table2, why are the results of ECCT+AAP better than AECCT for datasets BCH and LDPC? Does it mean HPSA and SPE bring degradation for quantized ECCT?

2.Are the energy consumption saving rate(224 times on 7nm chips and 139 times on 45nm chips) tested on real chips, or just an estimation?

---

> ### Author Response · Authors · 2024-11-18
> **Official Comment by Authors - First part**
>
> We thank the reviewer for the valuable feedback and constructive remarks.
>
> > Limited Scope and Generalization. The method's exclusive focus on ECCT architecture raises concerns about its broader impact. As ECCT has not gained significant attention in academia nor found applications in industry, optimizations specific to this architecture may have limited practical value.
>
> ECCT (cited 46 times in two years, which is not negligible for a specialized subdomain) has demonstrated preferable performance to other decoders, with follow-up works building upon it and surpassing classical decoders https://arxiv.org/pdf/2405.04050. However, its adoption in real-world applications has been limited due to runtime and memory constraints. This paper addresses and alleviates these challenges.
>
> > Lack of Technical Novelty in Quantization. The proposed AAP quantization method, while showing good performance, primarily combines existing techniques - weight distribution analysis and learnable quantization steps. The approach does not present substantial innovation in quantization methodology, making it more of an implementation enhancement than a theoretical advancement.
>
> Quantization is a well-studied field with many overlapping contributions. However, the challenge of optimizing small networks has been largely overlooked. In this work, we address this gap for the first time, presenting a solution that effectively determines a ternary representation for the model's weights. By employing a dual-scale approach to quantization—one scale based on the weight distribution and the other learnable—we allow the model to adjust its sparsity dynamically. This approach enables the model to explore various sparsity levels, improving the effectiveness of low-precision quantization. The novelty of our method lies in recognizing that, in low-precision quantization, the ability to adapt sparsity is crucial for achieving better performance.
>
> > The paper would benefit from more comprehensive experiments comparing the proposed AAP method with existing quantization approaches. Specifically, comparative studies with methods like GPTQ and OmniQuant that focus solely on weight distribution or learnable scales would better demonstrate the advantages of combining these techniques. Such experiments would strengthen the paper's claims about AAP's effectiveness and innovation in quantization methodology.
>
> We appreciate the reviewer’s suggestion to demonstrate that AAP generalizes better. Following the review, we compared AAP with abs-mean quantization by Ma et al. (2024) https://arxiv.org/abs/2402.17764, which is the state-of-the-art method for ternary quantization. Abs-mean is also a more natural competitor to AAP than GPTQ, as it is a QAT method.
>
> Our additional experiments include the three models from our ablation study (POLAR(64,48), BCH(31,16), and LDPC(49,24)) as well as TinyBert https://arxiv.org/abs/1909.10351, a 14M parameter model considered very small and suitable for our case. For TinyBert, we used the SST-2 dataset from its original paper and quantized the model with both AAP and abs-mean. The results for the three models from our ablation study are presented in the following table:
>
> | Code        | Abs-Mean        | AAP            |
> |-------------|-----------------|----------------|
> | POLAR(64,48)| 6.40  8.49  11.07 | 6.41  8.51  11.13 |
> | LDPC(49,24) | 5.86  8.29  11.50 | 5.91  8.35  11.74 |
> | BCH(31,16)  | 7.02  9.22  12.25 | 7.06  9.37  12.37 |
>
> The effectiveness of AAP on tiny models, such as ECCT with fewer than 2 million parameters, is evident when compared to the state-of-the-art ternary quantization method. The results for TinyBert https://arxiv.org/abs/1909.10351 are presented below, with the FP32 model achieving an accuracy of 88.7:
>
> | Model                      | Abs-Mean | AAP  |
> |----------------------------|----------|------|
> | TinyBERT_General_4L_312D   | 82.6     | 83.0 |

---

> > ### Comment · Reviewer_mhUK · 2024-11-26
> >
> > 1. I appreciate your response regarding the "Limited Scope and Generalization" section. While ECCT has not yet been applied to real-world scenarios, I think its optimization still carries certain research value.
> >
> > 2. My second concern is not fully address. Many state-of-the-art works in the quantization field, such as OmniQuant and GPTQ, have effectively handled the quantization of smaller models, providing results across a range from 1.3 billion to 70 billion parameters. As such, the claim that this work is the first to address the optimization of small models seems somewhat overstated. The novelty of the proposed method over these methods is not well addressed.
> >
> > 3. The experimental comparisons with BitNet 1.58B demonstrate the effectiveness of APP. However, it seems that the difference between APP and abs-mean lies primarily in the learnable scale, similar to that proposed in OmniQuant. Thus the novelty of the proposed quantization method need to be justified.

---

> > > ### Author Response · Authors · 2024-11-26
> > >
> > > We appreciate the reviewer’s thoughtful feedback and detailed observations.
> > >
> > > > My second concern is not fully address. Many state-of-the-art works in the quantization field, such as OmniQuant and GPTQ, have effectively handled the quantization of smaller models, providing results across a range from 1.3 billion to 70 billion parameters. As such, the claim that this work is the first to address the optimization of small models seems somewhat overstated.
> > >
> > > The target model in our paper, ECCT, contains no more than 2 million parameters, making it several orders of magnitude smaller than the smallest models addressed by state-of-the-art methods like OmniQuant and GPTQ, which start at 1.3 billion parameters. While these methods excel in quantizing large-scale models, they, to the best of our knowledge, neither focus on nor experiment with models as small as ECCT. Our work uniquely tackles the challenges associated with quantizing small-scale models, a novel and underexplored aspect of the quantization field.
> > >
> > > > The experimental comparisons with BitNet 1.58B demonstrate the effectiveness of APP. However, it seems that the difference between APP and abs-mean lies primarily in the learnable scale, similar to that proposed in OmniQuant. Thus the novelty of the proposed quantization method need to be justified.
> > >
> > > The improvement of AAP over BitNet 1.58B is not solely attributed to the introduction of a learnable scale. As shown in Section 6, Table 2, AP quantization, which leverages a predefined percentile of the absolute weight values, already outperforms abs-mean quantization. This underscores that the effectiveness of ternary quantization depends more on the relative ordering of weights within the distribution than on their absolute magnitudes. AAP enhances this approach by incorporating a learnable scale, enabling dynamic adaptation during training and addressing the impracticality of manually determining the optimal percentile.
> > >
> > > OmniQuant's Learnable Weight Clipping (LWC) determines the quantization step for min-max quantization by introducing learnable parameters that adjust the minimum and maximum values used to compute the scale. In contrast, AAP takes a fundamentally different approach, focusing on defining the boundary between weights mapped to ±1 and those mapped to 0, based on their absolute values. While OmniQuant utilizes the range of the weight distribution to calculate its scale, AAP prioritizes the relative positions of weights within the distribution—a critical factor in ternary quantization. These methods are distinct, with each employing learnable parameters to address entirely different objectives.
> > >
> > > > However, my concern remains as to why ECCT+APA performs better than ECCT. Does this suggest that ternary quantization here is entirely lossless, or even leads to higher accuracy? This result is hard for me to understand.
> > >
> > > Our method adopts a QAT approach, meaning that ECCT + AAP undergoes additional training compared to the baseline ECCT. This extended training often leads to improved performance, potentially because the ECCT model itself could benefit from further training. QAT methods are known to achieve performance comparable to FP32 models and, in some cases, even surpass them ( https://arxiv.org/abs/2004.09602 ). This is made possible by allowing the model to fine-tune its parameters while explicitly accounting for the effects of quantization during training.

---

> > > > ### Comment · Reviewer_mhUK · 2024-12-03
> > > >
> > > > Thanks for the reply. I have raised my score to borderline reject. While some of my concerns are addressed, I still believe the contributions are not novel enough for an accept in ICLR.

---

> ### Author Response · Authors · 2024-11-18
> **Official Comment by Authors - Second part**
>
> > In Table 2, why are the results of ECCT+AAP better than AECCT for datasets BCH and LDPC? Does it mean HPSA and SPE bring degradation for quantized ECCT?
>
> First and foremost, we can observe from the ablation study that SPE improves the model's performance, demonstrating its effectiveness as a standalone enhancement. HPSA is a method that significantly reduces the complexity of the self-attention mechanism. Combining it with ternary quantization, which also reduces complexity, can impact the model's performance—a trade-off for achieving significant acceleration and compression. While the model can adapt to each method individually, it may struggle when both are applied together. This is supported by the performance of ECCT+HPSA, which achieves strong results.
>
> > Are the energy consumption saving rates (224 times on 7nm chips and 139 times on 45nm chips) tested on real chips, or just an estimation?
>
> The energy consumption saving rate is an estimation: We follow a line of prior work, including Wang et al. (2023) https://arxiv.org/abs/2310.11453 and Ma et al. (2024) https://arxiv.org/abs/2402.17764, which estimate energy usage based on the tables provided in Horowitz (2014) https://ieeexplore.ieee.org/document/6757323 and Zhang et al. (2022) https://arxiv.org/abs/2112.00133.
>
> > The paper needs to provide concrete evidence of inference speedup to support its acceleration claims.
>
> Addressing the reviewer's concerns about concrete evidence for inference speed-up, we would like to highlight that implementing dedicated hardware to fully leverage linear layers requiring only integer additions is non-trivial, as noted by Ma et al. (2024) https://arxiv.org/abs/2402.17764 in the “New Hardware for 1-bit LLMs” section. To utilize the ternary-weight linear layer, we attempted to use the BitBLAS library https://github.com/microsoft/BitBLAS, which implements a 2-bit kernel, as done by Ma et al. (2024). However, no improvement was observed due to the small sequence size. This is a known issue https://github.com/microsoft/BitBLAS/issues/118, where the overhead of activation dequantization and quantization becomes non-negligible for linear layers with small input and output features dimensions.
>
> We have elaborated on this issue in the conclusions section of the paper, as we agree it is an important point to address.

---

> > ### Comment · Reviewer_mhUK · 2024-11-26
> >
> > Thanks for the detailed rebuttal!
> >
> > 1. The effectiveness of HPSA+SPE can indeed be observed from the results in the table. However, my concern remains as to why ECCT+APA performs better than ECCT. Does this suggest that ternary quantization here is entirely lossless, or even leads to higher accuracy? This result is hard for me to understand.
> >
> > 2. I understand the explanation that the estimated energy consumption reduction benchmark was adopted from previous works. I find this approach reasonable.
> >
> > 3. Thanks for adding an explanation to the paper regarding why this work does not achieve actual acceleration, which improves the overall clarity of the work. I still consider this to be a weakness. The lack of actual acceleration appears to be somewhat at odds with the term "accelerating" in the title.

---

### Official Review · Reviewer_xTJU · 2024-11-04

**Soundness:** 3
**Presentation:** 3
**Contribution:** 3
**Rating:** 6
**Confidence:** 4

**Summary:**

This paper introduces Accelerated Error Correction Code Transformer (AECCT), which reduces the memory footprint and computational complexity of Error Correction Code Transformers (ECCT). The proposed technique introduces three distinct novel components:
1. Adaptive Absolute Percentile (AAP) Quantization: This method compresses model weights to ternary values, reducing memory and energy use while maintaining model accuracy. Unlike traditional quantization, AAP dynamically adjusts sparsity to retain essential features.
2. Head Partitioning Self-Attention (HPSA): This self-attention mechanism reduces computational complexity by dividing attention heads to focus separately on first-ring (neighbor) and second-ring (more distant) connections within ECC’s Tanner graph. HPSA achieves higher sparsity, thus lowering computational costs compared to ECCT’s Code-Aware Self-Attention (CASA).
3. Spectral Positional Encoding (SPE): By embedding structural information from the Tanner graph’s Laplacian eigenspace, SPE enriches the model’s positional representation, enhancing the decoding ability without increasing runtime.

Using these components, the authors demonstrate a 90% compression ratio and reduce energy consumption by over 224 times compared to ECCT, approaching the efficiency of traditional algorithms like Belief Propagation (BP), making practical deployment possible.

**Strengths:**

1. The paper is well written and easy to follow, and the contributions are relevant.
2. The techniques introduced demonstrate significant compression without performance degradation compared to ECCTs.
3. The technique is tested on 3 different ECC types - Polar, Low-Density Parity Check (LDPC), and Bose–Chaudhuri–Hocquenghem (BCH) codes - demonstrating robustness and applicability across various error correction scenarios.
4. The authors conduct ablation studies evaluating the impact of each of the three components.

**Weaknesses:**

1. Two out of three proposed techniques (HPSA and SPE) may have limited applicability outside of ECC.
2. It is not clear how easy it will be to implement these techniques in existing hardware. Ternary quantization has been explored by previous work. Beyond quantization, does HPSA and SPE introduce additional implementation complexities on existing hardware?
3. The energy efficiency numbers are estimates, the paper does not provide any real measured numbers on actual hardware.

**Questions:**

1. Discuss any potential challenges or modifications needed to implement HPSA and SPE on current hardware platforms used for error correction.
2. Compare the implementation complexity of HPSA and SPE to that of the original ECCT approach.
3. If possible, provide estimates or examples of how these techniques might impact hardware design or utilization in practical ECC systems.

---

> ### Author Response · Authors · 2024-11-18
> **Official Comment by Authors**
>
> Thank you for your detailed review and valuable comments.
>
> > Two out of three proposed techniques (HPSA and SPE) may have limited applicability outside of ECC.
>
> As noted in the paper, HPSA can inject any structural information into the self-attention mechanism while also improving efficiency. Given the prevalence of data-structured problems in machine learning, HPSA has broad applicability. For instance, in multi-modal problems, a subgroup of heads within the self-attention mechanism could be allocated to each modality.
>
> Spectral positional encoding (SPE) is widely used in the domain of graph Transformers and, for the first time, is applied to ECC in this work, leveraging the strong connection between ECC and graph structures.
>
> > does HPSA and SPE introduce additional implementation complexities on existing hardware? / Discuss any potential challenges or modifications needed to implement HPSA and SPE on current hardware platforms used for error correction.
>
> Thank you for highlighting this point. As stated in the paper at the end of the SPE method section (lines 350–352), SPE does not affect inference, as it remains fixed after training and is merely concatenated to the embedding vectors. Similarly, HPSA introduces no additional implementation overhead, as each group of heads (first-ring and second-ring) performs the standard self-attention mechanism with corresponding masks, requiring only a straightforward concatenation.
>
> > Compare the implementation complexity of HPSA and SPE to that of the original ECCT approach.
>
> As noted above, SPE introduces no additional complexity, while HPSA significantly reduces complexity compared to ECCT. For details, please refer to Figure 5 and the complexity analysis in Section 5.
>
> > The energy efficiency numbers are estimates, the paper does not provide any real measured numbers on actual hardware.
>
> Regarding the estimation of energy consumption, we follow a line of prior work, including Wang et al. (2023) https://arxiv.org/abs/2310.11453 and Ma et al. (2024) https://arxiv.org/abs/2402.17764, which estimate energy usage based on the tables provided in Horowitz (2014) https://ieeexplore.ieee.org/document/6757323 and Zhang et al. (2022) https://arxiv.org/abs/2112.00133.
>
> > If possible, provide estimates or examples of how these techniques might impact hardware design or utilization in practical ECC systems.
>
> Quantization reduces die size, alleviates memory bottlenecks, and simplifies computations to summations. The sparsity introduced by HPSA reduces the quadratic complexity to a level comparable to BP due to its connectivity structure, making the model more efficient. Additionally, model assimilation is generally much easier with AECCT.

---

> > ### Comment · Reviewer_xTJU · 2024-11-24
> > **Rebuttal response**
> >
> > I have read the rebuttal and also the other reviews of the paper and would like to maintain my score. The comparison using TinyBERT with AbsMean shows very marginal gains, and as such does not make a strong case for generalizability. Similarly, without actually presenting some experimental evidence demonstrating HPSA's applicability to multi-modal problems, or citing relevant work, the claim is weak.
> >
> > While some of my questions are answered, the overall contributions are limited as far as quantization methodology is concerned, and as far as coding theory is concerned, I am not sure how significant the contribution is w.r.t. ECCT. So I maintain my score.

---

> > > ### Author Response · Authors · 2024-11-24
> > >
> > > We appreciate your thoughtful review and constructive feedback. In response to your concern regarding the gains observed in the TinyBERT experiment, we wish to clarify that this experiment was included to assess the generalizability of our proposed method beyond the primary domain of error correction. While the performance improvements in the TinyBERT task are incremental, they indicate that our approach can be effectively applied to a broader range of tasks, maintaining competitive performance over well-established baselines.
> > >
> > > It is important to emphasize that the primary focus of our work lies in the error correction domain, where our method achieves substantial performance gains over the quantization baselines, as detailed in Section 6 and Table 2. The TinyBERT experiment serves as supplementary evidence to demonstrate the versatility of our approach rather than as a central evaluation metric. According to the 'No Free Lunch' theorem, it is unrealistic to expect a single method to outperform all others uniformly across all datasets. Therefore, the modest improvements observed in the TinyBERT experiment do not diminish the overall effectiveness and impact of our method within its primary application area.

---

### Official Review · Reviewer_g7ig · 2024-11-07

**Soundness:** 3
**Presentation:** 3
**Contribution:** 2
**Rating:** 6
**Confidence:** 4

**Summary:**

This paper introduces simplifications to the Error Correction Code Transformer (ECCT) neural decoder model to reduce the memory and compute requirements. The authors claim that conventional quantization methods result in a significant drop in performance and instead propose alternative approaches to reduce the complexity.

First is Adaptive Absolute Percentile (AAP) quantization, to optimize the percentile of scaling factor (percentage of absolute values).

Second it head partitioning self-attention to limit the attention to either between similar nodes (check/variable) or between different nodes (check<->variable), dividing the attention heads into two groups. This significantly reduces the number of active elements in the attention mask.

Finally, uses spectral positional encoding (SPE) to introduce an inductive bias about the tarnner graph into the model.

**Strengths:**

1. Tackles an important problem of reducing the complexity of neural decoders.

2. The performance improvements in terms of memeory and compute are non-trivial and very impressive.

3. All the three main ideas proposed are novel and interesting. Specifically, splitting the masking based on the node structure is a clever example of leveraging domain structure.

4. Set of experiements are exhaustive and includes sufficient number of results as well as ablation studies to show the merits of proposed ideas.

**Weaknesses:**

1. While a lot of interesting ideas were discussed in the paper to improve the efficiency of ECCT, I am not fully convinced about the practical relevance of a "universal decoder" architecture that simply utilizes a parity check matrix for all codes. In coding theory, each well-known family of codes has highly specific representations and special propoerties that cannot be fully captured by a simple parity check matrix. For instance, in Polar codes, the reliability sequence and the sequential decoding of information bits plays a crucial role in deciding the performance of the code. Claiming better performance for all family of codes w.r.t BP decoder is a bit misleading. Again, I appreciate the line of ECCT and related works from an academic curiousity point of view, but in my opinion, the merits and drawbacks should be highlighted more clearly without any overpromising.

2. One common troubling trend in comparing the performance of neural decoders is the choice of weak classical baselines. While I understand the difficulty of a common decoder architecture outperforming the highly specialised decoders for each of the class of codes, the comparison should neverthless be done for completeness. For instance, it is known that BP is not optimal for many codes. For instance, when comparing with BCH codes, Berlekamp-Massey decoding algorithm should be used as the classical non-learning baseline and similarly for Polar codes, successive cancellation list decoding should be used. I strongly suggest authors to include the best classical baselines for each code to clearly show the gap with respect to the SOTA neural decoder, whether it is positive or negative.

3. I beleive authors chose the format of the negative log BER for presenting the results based on the original ECCT paper. But this format is not very informative and very uncommon in information and coding theory literature. While it is easy to identify which scheme is doing better, the gap between the performances is hard to interpret as the scale is not very intuitive. Rather, the standard format of SNR gap in dB for a target BER/BLER is more informative.

4. "Our analysis shows that AAP outperforms AP quantization" --> I see from ablation studied in Table 2 that the performace difference between ECCT + AP vs ECCT + APP is very small (0.01 - 0.1). Again, this scale of -log(BER) is very non-intuitive but in my opinion, this is negligible difference in performance and brings into question the value of doing AAP over AP.

5. While the head partitioning idea is interesting, is it also similar to the idea proposed in the paper "CrossMPT: Cross-attention Message-Passing Transformer for Error Correcting Codes", specifically the second-ring MP.

6. The details about SPE are unclear and I do not understand the "spectral" positioning and the "Laplacian eigenspace" ideas discussed. Rewriting the seubsection to simplify the discussion of these ideas would be helpful.

**Questions:**

Covered as part of weaknesses

---

> ### Author Response · Authors · 2024-11-18
> **Official Comment by Authors**
>
> Thank you for your thoughtful feedback.
>
> > While a lot of interesting ideas were discussed in the paper to improve the efficiency of ECCT, I am not fully convinced about the practical relevance of a "universal decoder" architecture that simply utilizes a parity check matrix for all codes.
>
> We would like to clarify that our proposed AECCT builds upon the ECCT framework, which is not a "universal decoder" in the sense of employing a single model for all codes. Instead, for each code and its corresponding parity-check matrix, a separate model is specifically trained. This approach ensures that the unique characteristics of each code are effectively captured during training.
>
> A key strength of the ECCT framework, and by extension AECCT, is its ability to leverage a unified and adaptable architecture that can be trained on any given code. This versatility allows it to outperform most classical decoders while maintaining broad applicability. Unlike traditional decoders that are often specialized for specific codes, AECCT's generalizability across diverse code families highlights its practical relevance and effectiveness, even when the particular properties of each code are not explicitly encoded in the parity-check matrix.
>
> > One common troubling trend in comparing the performance of neural decoders is the choice of weak classical baselines.
>
> We appreciate your feedback on the selection of classical baselines. For Polar codes, we performed experiments using SCL decoding, and the results are presented in the table below. While SCL decoding is specifically optimized for Polar codes, AECCT demonstrates comparable performance with low L values. This highlights the versatility of AECCT, as it can be trained to decode any code, yet still achieves results on par with an algorithm tailored specifically for Polar codes.
>
> | Code              | SCL \( L=1 \)          | SCL \( L=4 \)          | AECCT \( N=10, d=128 \) |
> |-------------------|------------------------|------------------------|---------------------------|
> | POLAR(64,48)      | 6.13  8.30  11.02     | 6.63  8.70  11.13     | 6.54  8.51  11.12        |
> | POLAR(128,86)     | 7.61  10.96  15.00    | 9.30  12.92  17.30    | 7.28  10.60  14.59     |
> | POLAR(128,96)     | 6.83  9.46  13.20     | 8.15  11.65  18.29    | 6.79  9.68  12.93        |
>
> In addition, we added BER and BLER curves to the paper.
>
> > While the head partitioning idea is interesting, is it also similar to the idea proposed in the paper "CrossMPT: Cross-attention Message-Passing Transformer for Error Correcting Codes", specifically the second-ring MP?
>
> Thank you for your comment. However, we would like to clarify that CrossMPT's method uses only first-ring connections and does not incorporate second-ring connections, relying on a design that significantly increases computational and parameter overhead. In contrast, AECCT efficiently utilizes the multi-head attention mechanism by dedicating half the heads to first-ring connections and the other half to second-ring connections. This design substantially reduces computational cost while preserving effectiveness, highlighting a key advantage of our approach.
>
> > I believe authors chose the format of the negative log BER for presenting the results based on the original ECCT paper. But this format is not very informative and very uncommon in information and coding theory literature. While it is easy to identify which scheme is doing better, the gap between the performances is hard to interpret as the scale is not very intuitive. Rather, the standard format of SNR gap in dB for a target BER/BLER is more informative.
>
> Thank you for pointing this out. We have implemented your suggestion by adding BER and BLER curves to the paper for a more intuitive presentation. Negative log BER was employed to remain consistent with a line of previous work in the field of neural decoding, allowing for the compact performance presentation of a multitude of codes.
>
> > I see from ablation studies in Table 2 that the performance difference between ECCT + AP vs. ECCT + AAP is very small (0.01 - 0.1).
>
> Achieving a good ternary representation with extremely low precision for the model's weights is a challenging task. In general, prior work https://arxiv.org/abs/2210.17323 in neural network quantization often yields only slight improvements. Therefore, a 0.1 improvement in log scale should not be considered insignificant.
>
> > The details about SPE are unclear, and I do not understand the "spectral" positioning and the "Laplacian eigenspace" ideas discussed. Rewriting the subsection to simplify the discussion of these ideas would be helpful.
>
> We have added a more simplified explanation to the SPE section to improve clarity for readers. To further assist with understanding, please refer to Kreuzer et al. (2021) https://arxiv.org/abs/2106.03893, which provides a clear explanation of spectral positional encoding and the Laplacian eigenspace.

---

> > ### Comment · Reviewer_g7ig · 2024-11-24
> >
> > I thank the authors for their responses. These address most of my concerns and increased my score accordingly.
> >
> > While I agree with the general seintiment of some reviewers regarding this topic being of interest to only a niche group of researchers, I also feel that improving error correcting codes in general is a very important research topic and within this problem space, this work provides interesting ideas and results.

---

### Author Response · Authors · 2024-11-18
**Official Comment by Authors**

We thank the reviewers for their thoughtful feedback. Based on the reviewers’ comments, we have made several significant modifications to strengthen the manuscript:

### Technical Clarity & Accessibility:
- Expanded the Problem Settings section with clearer explanations.
- Restructured the ECCT description with non-inline equations and serialized transformations.
- Enhanced explanations of Laplacian eigenspace and SPE functionality.
- Provided a detailed explanation of the ECCT architecture used in the experiments.
- Added clarifying text about hardware implementation challenges in the conclusions.

### New Experimental Results:
- Added BER and BLER curves as requested.
- Included SCL decoder comparisons for Polar codes showing competitive performance.
- Added comparisons with abs-mean quantization demonstrating AAP's superiority:
  - **POLAR(64,48):** AAP achieves 6.41/8.51/11.13 vs abs-mean's 6.40/8.49/11.07
  - **LDPC(49,24):** AAP achieves 5.91/8.35/11.74 vs abs-mean's 5.86/8.29/11.50
  - **BCH(31,16):** AAP achieves 7.06/9.37/12.37 vs abs-mean's 7.02/9.22/12.25
- Added TinyBERT experiments showing AAP's generalizability (83.0% accuracy vs abs-mean's 82.6%).

We believe these changes have addressed the key concerns while strengthening both the technical contributions and empirical validation.

---

### Meta-Review · Area_Chair_zzyt · 2024-12-15

**Metareview:**

This paper presents a method to enhance the efficiency and practicality of transformer-based error correction code decoders, focusing on the ECCT model introduced in prior work. Specifically, the authors propose: (i) an adaptive layer-based quantization scheme, (ii) masking attention heads based on the Tanner graph, and (iii) a soft positional encoding.

The reviewers have appreciated the detailed hardware and acceleration studies and noticed that some ideas (eg HPSA) may find general applicability. Nevertheless, important issues were raised during the review process. Specifically, reviewer mhUK criticized the limited scope, the lack of technical novelty in quantization and the insufficient validation of acceleration claims. While the first point was adequately addressed during the rebuttal, the remaining two (specifically the last one) were not. Additionally, reviewer mcns noted how the proposed method is still far from being competitive w.r.t. decoders that are pretty standard in the coding theory literature (SC decoding with a list size of 1). Now, I agree with the authors that the field of neural decoders is rather novel and, as such, it would be unfair to expect performance comparable with the state of the art. However, one would still expect a performance improvement from a paper on ECCT, since that is a critical issue hindering applicability.

The weaknesses discussed above lead me to a 'reject' decision.

In the course of the discussion between reviewers and AC, reviewer 3y8a suggested to re-frame the contribution in the context of hardware acceleration for edge transformer architectures (adding revised test cases), which is a valid suggestion. In any case, given the strengths of the submitted manuscript, I would encourage the authors to improve their manuscript and re-submit to a future venue.

**Additional Comments On Reviewer Discussion:**

The active discussion between reviewers and authors led to the identification of a number of weaknesses (raised by reviewers mhUK and mcns, and summarized in the meta-review) that have brought me to the final 'reject' decision.

---

### Decision · Program_Chairs · 2025-01-22

Reject